# Atypical collective oscillatory activity in cardiac tissue uncovered by optogenetics

**Alexander S Teplenin[1]\*, Nina N Kudryashova[1‡], Rupamanjari Majumder[1§], Antoine AF de Vries[1], Alexander V Panfilov[1,2], Daniël A Pijnappels[1]\*[†], Tim De Coster[1,3]\*[†]**

[1]Laboratory of Experimental Cardiology, Department of Cardiology, Heart Lung Center Leiden, Leiden University Medical Center, Leiden, Netherlands; [2]Department of Physics and Astronomy, Ghent University, Ghent, Belgium; [3]Netherlands Heart Institute, Utrecht, Netherlands

\*For correspondence:
a.teplenin@lumc.nl (AST);
d.a.pijnappels@lumc.nl (DAP);
t.j.c.de_coster@lumc.nl (TDC)

[†]These authors contributed equally to this work

**Present address:** [‡]Institute for Adaptive and Neural Computation, Informatics Forum, School of Informatics, the University of Edinburgh, Edinburgh, United Kingdom; [§]Nantes Université, INSERM, l'institut Du Thorax, Nantes, France

**Competing interest:** The authors declare that no competing interests exist.

## eLife Assessment

This **important** work provides mechanistic insights into the development of cardiac arrhythmia and establishes a new experimental use case for optogenetics in studying cardiac electrophysiology. The agreement between computational models and experimental observations provides a **convincing** level of evidence that wave train-induced pacemaker activity can originate in continuously depolarized tissue, with the limitation that there may be differences between depolarization arising from constant optogenetic stimulation, as opposed to pathophysiological tissue depolarization. Future experiments in vivo and in other tissue preparations would extend the generality of these findings.

**Abstract** Many biological processes emerge as frequency-dependent responses to trains of external stimuli. Heart rhythm disturbances, that is cardiac arrhythmias, are important examples as they are often triggered by specific patterns of preceding stimuli. In this study, we investigated how ectopic arrhythmias can be induced by external stimuli in cardiac tissue containing a localised area of depolarisation. Using optogenetic in vitro experiments and in silico modelling, we systematically explored the dynamics of these arrhythmias, which are characterised by local oscillatory activity, by gradually altering the degree of depolarisation in a predefined region. Our findings reveal a bi-stable system, in which transitions between oscillatory ectopic activity and a quiescent state can be precisely controlled, that is by adjusting the number and frequency of propagating waves through the depolarised area oscillations could be turned on or off. These frequency-dependent responses arise from collective mechanisms involving stable, non-self-oscillatory cells, contrasting with the typical role of self-oscillations in individual units within biophysical systems. To further generalise these findings, we demonstrated similar frequency selectivity and bi-stability in a simplified reaction–diffusion model. This suggests that complex ionic cell dynamics are not required to reproduce these effects; rather, simpler non-linear systems can replicate similar behaviour, potentially extending beyond the cardiac context.

## Introduction

Biological systems like the heart typically display complex dynamical behaviour through interactions among components such as genes, proteins, or cells (*Dana et al., 2008*). These systems exhibit non-linear dynamics, feedback loops, and adaptability, reflecting phenomena like homeostasis, evolution, and rhythm stability (*Xiong and Garfinkel, 2023*). Central to the framework of dynamical systems is the concept of the phase space, a multidimensional representation in which each point corresponds

to a unique state of the system. In geography, when a small ball rolls on a raised-relief map, a state would include the planar position, elevation, speed, and acceleration of the ball.

Within the phase space, the long-term behaviour of a system can often be described in terms of attractors, which are subsets of the phase space that trajectories tend towards over time. Attractors can take various forms, including fixed points, periodic orbits, and chaotic sets, reflecting the rich diversity of behaviours possible in dynamical systems. By replacing the ball with local rain in the raised-relief map, rivers, lakes, and other bodies of water are attractors where that rain is collected. A watershed, also known as a drainage basin, is an area of land where any rain that falls will drain into the same river, lake, or body of water. Therefore, they are basins of attraction. Mountain ridges are sepa-ratrices that divide the watersheds, or more formally theoretical surfaces that divide the phase space into different basins of attraction. A separatrix therefore denotes the transition between different regions of the phase space. The terminology introduced here comes back throughout our paper, with the addition of a 'state', that is one of the attractors in the system. When the system is not interacted with, it will stay at the same point(s) in phase space.

Some systems only have a single basin of attraction. However, in others, the phenomenon of bi-sta-bility emerges, where two distinct attractors coexist, each with its own basin of attraction. Bi-stability is a hallmark of many natural and engineered systems, from climate dynamics to neural networks, ecological models, and cardiac dynamics (*Angeli et al., 2004*). The interplay between basins in such systems can give rise to complex behaviours, making the characterisation of these basins a critical component of a system's analysis.

In this manuscript, the focus is on the complex system of the heart. At rest, ventricular cardiomyo-cytes (i.e muscle cells of the lower heart chamber) have a membrane potential of –80 to –90 mV. This resting state is an attractor of the system. When pushed out of equilibrium (e.g. due to an influx of $Na^+$ ions or electrical stimulation), cardiomyocytes first depolarise and subsequently get attracted again towards the resting state by an efflux of $K^+$ ions. In phase space, the trajectory that gets mapped out resembles a cycle, corresponding to one heartbeat. When multiple of these cells are connected to each other, emergent behaviour can arise. Within this dynamic system of cardiomyocytes, we investi-gated emergent bi-stability (a concept that will be explained more thoroughly later on) in cell mono-layers under the influence of spatial depolarisation patterns. By using a train of external (electrical) waves, the system could be actively moved from one basin of attraction to another.

Cardiac arrhythmias are typically initiated by specific triggers, with point-source waves, that is ectopic waves, being the most prevalent. To study ectopy in laboratory settings, control needs to be exerted over the depolarisation of cardiomyocytes. This can be achieved by optogenetics, that is a biological technique to control the activity of cells with light. This approach relies on the introduc-tion of recombinant genes encoding light-activatable proteins into cells to endow them with a new biological function (*Tan et al., 2022*). One way to depolarise a cardiomyocyte is by using a mini-singlet oxygen generator (miniSOG). Upon activation with blue light, miniSOG converts molecular oxygen from its ground state ($^3O_2$) into highly reactive singlet oxygen ($^1O_2$). The resulting oxidative damage induces a state of prolonged depolarisation in cardiomyocytes following electrical stimulation (*Jang-sangthong et al., 2016*). In such a system, it was previously observed that spatiotemporal illumina-tion can give rise to collective behaviour and ectopic waves (*Teplenin et al., 2018*) originating from illuminated/depolarised regions (with spatial high curvature). These waves resulted from the interplay between the diffusion current (also known in biology/biophysics as the gap junction mediated current) and the single-cell bi-stable state (*Teplenin et al., 2018*) that was induced in the illuminated region.

Although effects of illuminating miniSOG with light might lead to formation of depolarised areas, it is difficult to control the process precisely since it depolarises cardiomyocytes indirectly. Therefore, in this manuscript, we used light-sensitive ion channels to obtain more refined control over cardio-myocyte depolarisation. These ion channels allow the cells to respond to specific wavelengths of light, facilitating direct depolarisation (*Ördög et al., 2023*; *Ördög et al., 2021*). By inducing cardiomyo-cyte depolarisation or hyperpolarisation only in the illuminated areas, optogenetics enables precise spatiotemporal control of cardiac excitability, an attribute we exploit in this manuscript (*Figure 1—figure supplement 1*). With the aid of the light-gated ion channel CheRiff (*Hochbaum et al., 2014*), ectopic waves similar to those obtained in the miniSOG system could be elicited (*Figure 1—figure supplement 2*). Due to the superior control over depolarisation imposed by CheRiff, we uncovered new effects and unobserved behaviour related to the transition from a monostable cardiac system to

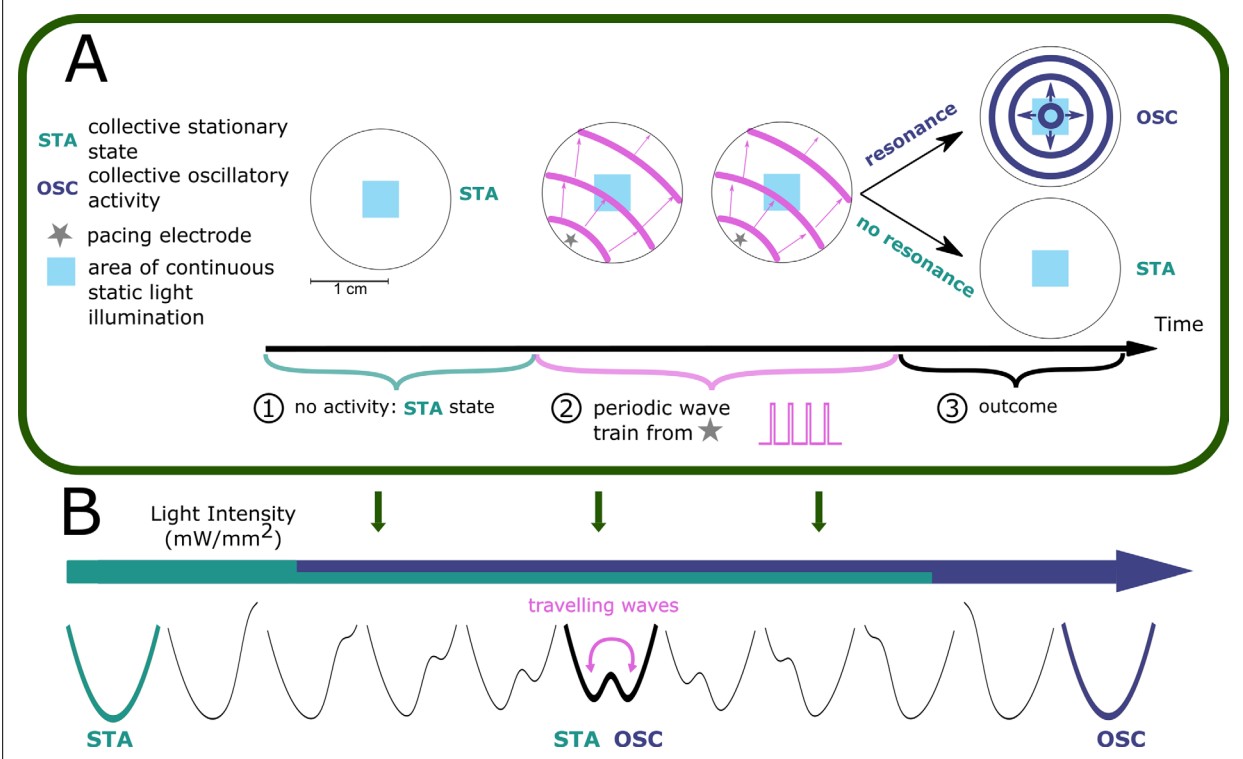

**Figure 1.** Bi-stability between stationary behaviour and collective oscillations. (**A**) Experimental timeline for a monolayer of optogenetically modified neonatal rat ventricular myocytes (NRVMs) under constant illumination of its centre: (1) observation of a collective stationary state (STA); (2) periodic wave train from the pacing electrode; and (3) outcome observation of collective behaviour, either stationary (STA) or oscillatory (OSC). (**B**) Schematic representation showing how light intensity influences collective behaviour of excitable systems, transitioning between a stationary state (STA) at low illumination intensities and an oscillatory state (OSC) at high illumination intensities. Bi-stability occurs at intermediate light intensities, where transitions between states are dependent on periodic wave train properties. TR.OSC, transient oscillations.

The online version of this article includes the following video and figure supplement(s) for figure 1:

**Figure supplement 1.** Experimental setup.

**Figure supplement 2.** Optogenetic depolarising tools and ectopy.

**Figure supplement 3.** A range of collective phenomena is observed in a multicellular in silico neonatal rat ventricular myocyte (NRVM) model.

**Figure 1—video 1.** A range of collective phenomena is observed in a multicellular in silico neonatal rat ventricular myocyte (NRVM) model.

https://elifesciences.org/articles/107072/figures#fig1video1

a bi-stable one that displays periodic oscillations. This transition resulted from the interaction of depolarised cardiomyocytes in an illuminated region with an external wave train not originating from within the illuminated region. More specifically, we found that our system displayed frequency selectivity: it oscillated only as a reaction to a specific range of inter-pulse external stimuli times. In neuroscience, this behaviour is classified as 'resonance'. However, to avoid confusion with classical resonance, where the strength of the response is dependent on the input, here the phenomenon is called 'induced pacemaker activity'. Evidence for this induced pacemaker activity will be presented both in vitro and in silico.

## Results

Bi-stability in cardiomyocyte monolayers was realised both in vitro and in silico (*Figure 1*). In vitro, this happened through patterned illumination applied to neonatal rat ventricular myocyte (NRVM) monolayers, which were optogenetically modified using the blue light-activatable cation channel CheRiff (*Figure 1—figure supplement 1*). In silico, use was made of a slightly modified (see Materials and methods) NRVM computational model (*Majumder et al., 2016*) complemented with a model for a depolarising light-gated ion channel (*Williams et al., 2013*). In both cases, the illumination pattern

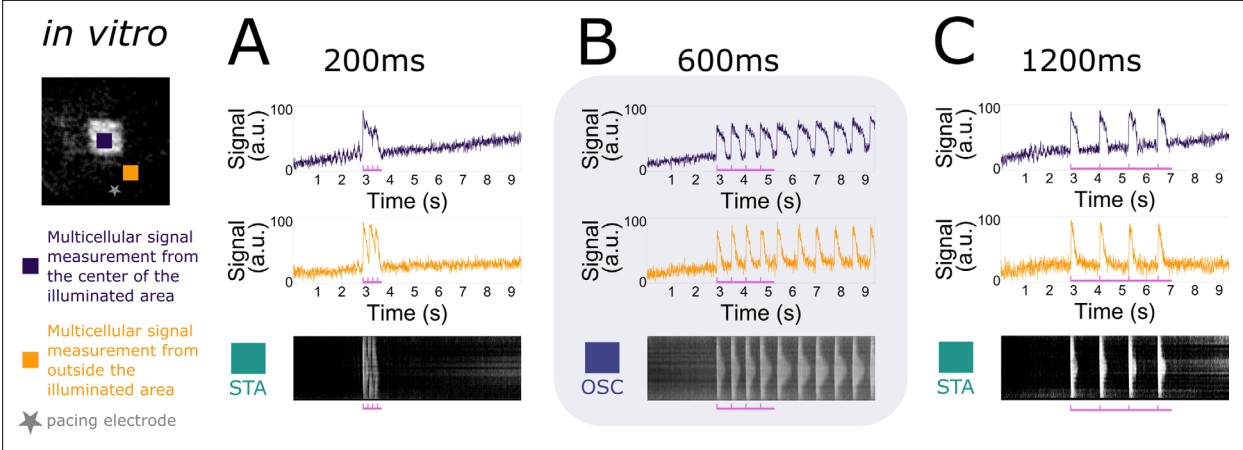

**Figure 2.** Frequency dependency of induced collective pacemaker activity. In vitro monolayers of optogenetically modified neonatal rat ventricular myocytes (NRVMs) show that the transition from the stationary state (STA) towards the oscillatory state (OSC) is dependent on the frequency of excitatory waves passing through an illuminated area under a constant number of four pulses. (**A**) Four pulses at high frequency (5.000 Hz or a 200-ms period) are not enough to induce the transition. (**B**) Four pulses at medium frequency (1.666 Hz or a 600-ms period) are sufficient to induce the transition. (**C**) Four pulses at low frequency (0.833 Hz or a 1200-ms period) fail once again in inducing the transition.

consisted of a rectangle of variable light intensities. After initiating a periodic wave train (of variable pulse number and interpulse interval/period) from a pacing electrode, two main outcomes and one special case were observed (*Figure 1A*). The first outcome was that the tissue stayed in the stationary rest state (STA). The second outcome was that the illuminated region became a pacemaker, which sent out pulses from its edges as a form of oscillatory behaviour (OSC). The special case consisted of an in-between case, in which the illuminated region only sends out a limited amount of pulses before returning to the stationary state, thus showing transient oscillatory behaviour (TR.OSC). Except for the occasional mention, this last behaviour of transient oscillations will be reserved for more detailed discussion in the supplement. All light intensity-influenced monolayer behaviours are visualised together in *Figure 1—figure supplement 3* and the accompanying *Figure 1—video 1*. Here, we will focus on the two main behaviours after periodic wave trains: STA and OSC. As shown below, these two behaviours can exist within the same monolayer under bi-stability and transitions from one state to the other are possible using periodic wave trains. The occurrence of this phenomenon appears to be influenced by the light intensity applied to the illuminated region (*Figure 1B*). In a step-by-step manner, we unravelled the intricacies of this tissue level (i.e. collective) bi-stability between oscillatory and stationary behaviour, linking experiment to theory.

## In vitro realisation of collective light-induced bi-stability

To test the scope of collective bi-stability in vitro, multiple parameters were varied within our experimental setup. The first one of these was the frequency of stimulation, visualised in *Figure 2*, where the number of electrical pulses and the light intensity (in the range 0.03125–0.25 mW/mm$^2$) were kept fixed. In every NRVM monolayer (26 biological replicates/monolayers), three frequencies (5.000, 1.666, and 0.833 Hz or stimulation periods 200, 600, and 1200 ms) were tested subsequently, with a total of three repetitions (technical replicates) of the full set, totalling $26 \times 3 = 78$ observations at each pacing frequency. Optical voltage signals were measured both at the centre of the illuminated area (top traces) as well as in the bulk of the tissue (bottom traces), in order to visualise action potentials throughout the tissue. Underneath these optical voltage traces is a space–time plot of the experimental monolayer, the vertical axis showing space and the horizontal one showing time. At each moment in time, the middle line of the monolayer image was plotted. White shows depolarised tissue, while black signifies repolarised cells. From these plots, intermediate stimulation periods (600 ms) were observed to change the monolayer from a collective stationary state to a collective oscillatory state, while low (200 ms) or high (1200 ms) ones had no effect. It was observed that the oscillatory state consisted of ectopic pacemaker activity with a frequency (1.361 Hz or a 735-ms period) that was different from the stimulus train frequency (1.666 Hz or a 600-ms period). Similar results were found

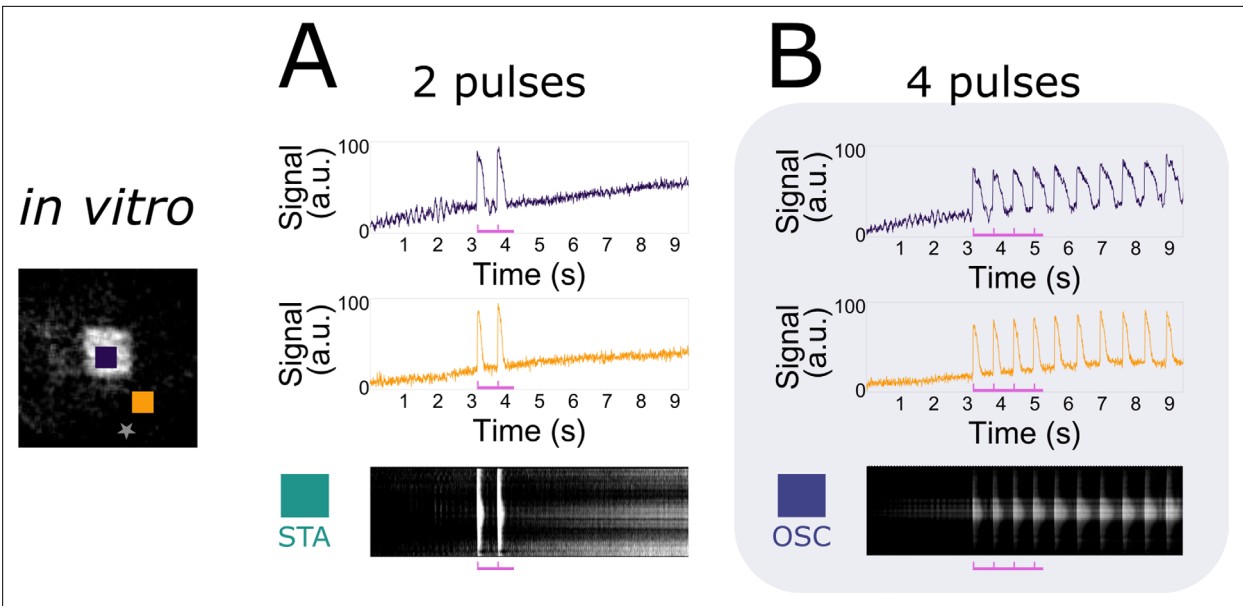

**Figure 3.** Accumulation of pulses to induce collective pacemaker activity. In vitro monolayers of optogenetically modified neonatal rat ventricular myocytes (NRVMs) show that the transition from the stationary state (STA) towards the oscillatory state (OSC) is dependent on the number of excitatory waves passing through an illuminated area at a constant electrical pacing frequency of 1.666 Hz or a 600-ms period. (**A**) Two pulses are not enough to induce the transition. (**B**) Four pulses are sufficient to induce the transition.

for the transient oscillatory state (*Figure 4—figure supplement 1A*). In short, we found that collective bi-stability becomes apparent in the mid-range of electrical pacing frequencies.

A second parameter tested was the number of pulses, visualised in *Figure 3*, where the period of stimulation (600 ms) and the light intensity (in the range 0.03125–0.25 mW/mm$^2$) were kept fixed. In every NRVM monolayer (7 biological replicates/monolayers), two different pulse numbers (2 and 4) were tested subsequently, with a total of three repetitions (technical replicates) of the full set, totalling 7 × 3 = 21 observations for each number of pulses. Once again, optical voltage traces were taken from the centre of the illuminated area and from the bulk of the tissue. Our observations show that a minimum number of pulses is needed to change the collective behaviour of the monolayer from stationary to oscillatory.

Taken together, we found a way to control transitions of collective behaviour of cardiomyocyte monolayers from stationary to oscillatory through two different parameters, that is the frequency and number of pulses.

## In silico realisation of collective light-induced bi-stability

Using the power of state-of-the-art computational modelling, we performed similar experiments to the in vitro ones with a modified version of the *Majumder et al., 2016* model for NRVMs combined with the *Williams et al., 2013* model for the H134R mutant of *Chlamydomonas reinhardtii* channelrhodopsin-2. As this light-sensitive ion channel slightly differs from CheRiff in its biophysical properties, the in vitro findings will have a qualitative rather than a quantitative correspondence to the in silico results, which are visualised in *Figure 4*. Simulating at a fixed constant illumination (1.7124 mW/mm$^2$) and a fixed number of four pulses, frequency dependency of collective bi-stability was reproduced in *Figure 4A*. Similar to the experimental observations, only intermediate electrical pacing frequencies (2.000 Hz or a 500-ms period) caused transitions from collective stationary behaviour to collective oscillatory behaviour and ectopic pacemaker activity had periods (710 ms) that were different from the stimulation train period (500 ms). *Figure 4B* shows the accumulation of pulses necessary to invoke a transition from the collective stationary state to the collective oscillatory state at a fixed stimulation period (600 ms). Also in the in silico simulations, ectopic pacemaker activity had periods (750 ms) that were different from the stimulation train period (600 ms). Also for the transient oscillatory state, the simulations show frequency selectivity (*Figure 4—figure supplement 1B*).

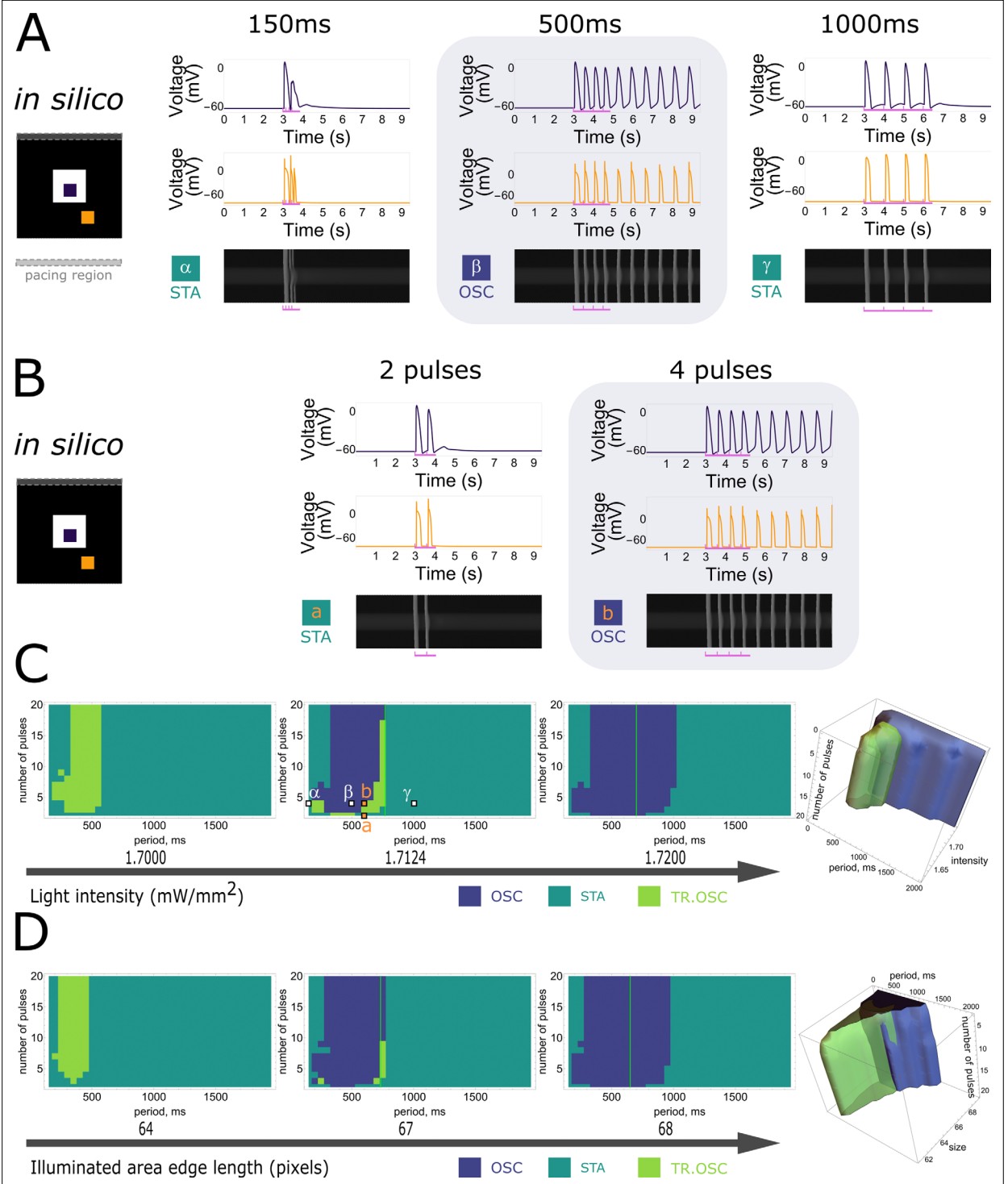

**Figure 4.** In silico demonstration of frequency dependency and accumulation of pulses to induce collective pacemaker activity similar to that observed in vitro. (**A**) Monolayers of optogenetically modified neonatal rat ventricular myocytes (NRVMs) show that the transition from the stationary state (STA) towards the oscillatory state (OSC) is dependent on the frequency of excitatory waves passing through an illuminated area under a constant number of four pulses. From left to right: no induction at stimulation periods of 150 and 1000 ms, induction at a stimulation period of 500 ms. (**B**) Monolayers of optogenetically modified NRVMs show that the transition from the stationary state towards the oscillatory state is dependent on the number of excitatory waves passing through an illuminated area under a constant frequency (1.666 Hz or a 600-ms period). From left to right: no induction at 2 pulses, induction at 4 pulses. (**C**) Three-dimensional parameter scan of all variables (light intensity, pacing period, and number of pulses) showing how dependency on period and on number of pulses relate to each other. Representative slices for fixed light intensities are displayed at the left. Vertical

*Figure 4 continued on next page*

*Figure 4 continued*

green lines show the natural pacemaker frequency the monolayer settles to after initiation of pacemaker activity. Letters in the middle panel correspond to traces in (A) and (B). (**D**) Three-dimensional parameter diagram in which the size of the illuminated area was varied instead of the light intensity. Representative slices for fixed area edge lengths are displayed at the left. Vertical green lines show the natural pacemaker frequency the monolayer settles to after initiation of pacemaker activity.

The online version of this article includes the following figure supplement(s) for figure 4:

**Figure supplement 1.** Frequency dependency of transient pacemaker activity induction.

**Figure supplement 2.** Influence of the size of the illuminated central square on transient pacemaker activity induction.

Having established a qualitative correspondence between the in silico model and in vitro experiments, the unique properties of the computational model were exploited to assert control over all variables in order to determine the parameter ranges of the state transitions. We previously identified three important parameters influencing such transitions: light intensity, number of pulses, and frequency of pulses. Scanning through all three variables revealed the specific conditions at which transition from the stationary state towards the oscillatory state occurred, as shown in the rightmost panel of *Figure 4C* with slices for fixed illumination extracted from this 3D plot shown at the left. With increasing light intensity, the parameter range (pulse and period) in which bi-stability can be detected increases (the total area of TR.OSC + OSC becomes larger), and transient oscillatory states transition towards oscillatory states.

Except for the three aforementioned parameters, a fourth one was found to have an influence on the state transitions: the size of the illuminated region. Also, this parameter was first tested in vitro, resulting in 5 biological replicates/monolayers showing transitions to the oscillatory state and 6 biological replicates/monolayers showing transient oscillations. The effect of the size of the illuminated area on the occurrence of bi-stability was qualitatively reproduced in silico (*Figure 4—figure supplement 2*). Varying the number and frequency of pulses together with the illuminated area at a fixed light intensity of 1.7200 mW/mm² resulted in the dynamic behaviour shown in *Figure 4D*. For the conditions that resulted in stable oscillations, the green vertical lines in the middle and right slices represent the natural pacemaker frequency in the oscillatory state. After the transition from the stationary towards the oscillatory state, oscillatory pulses emerging from the illuminated region gradually dampen and stabilise at this period, corresponding to the natural pacemaker frequency. The location of this line depends on the light intensity and the size of the illuminated region. However, it is independent of the number of pulses and the frequency of the stimulus train. *Figure 4C, D* also

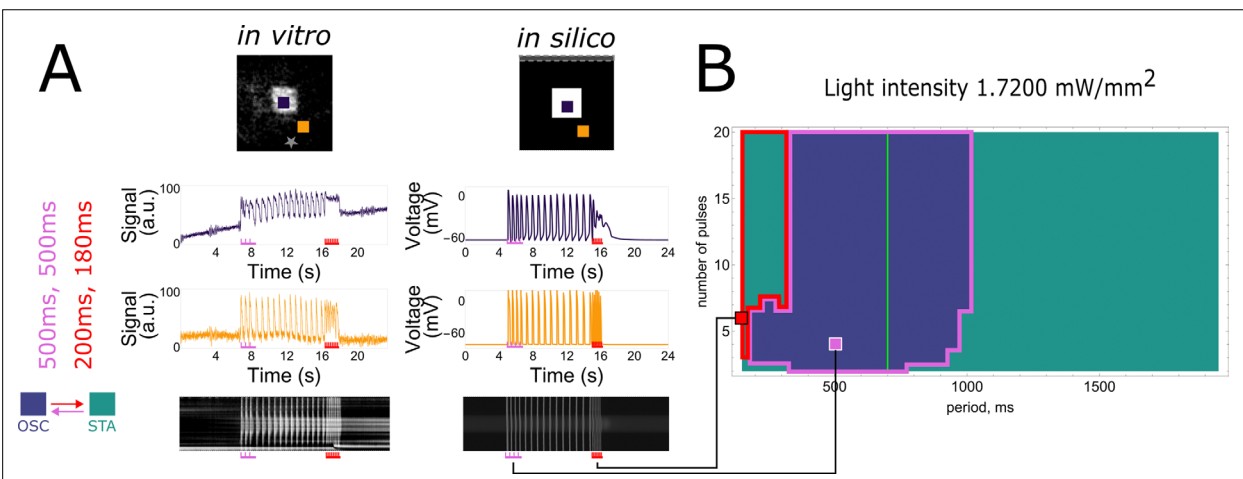

**Figure 5.** Demonstration of frequency dependency to terminate collective pacemaker activity. (**A**) Initiation and termination of collective pacemaker activity in vitro (left panel, three pulses of 500 ms period for initiation, eight pulses of 200 ms period for termination) and in silico (right panel, four pulses of 500 ms period for initiation, seven pulses of 180 ms period for termination). (**B**) Rightmost slice from *Figure 4C* showing in silico experiments at a fixed light intensity (1.72 mW/mm²) and size of the illuminated area (67 pixels edge length) with indicated termination (red border) and initiation (magenta border) period ranges. A vertical green line shows the natural pacemaker frequency the monolayer settles to after initiation of pacemaker activity.

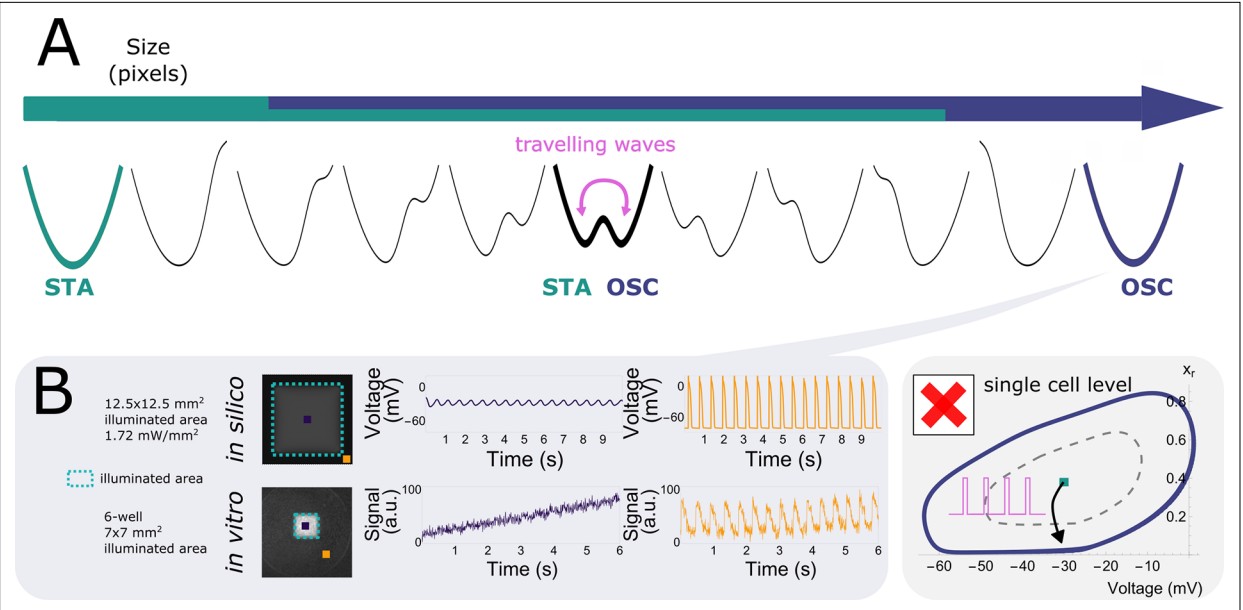

**Figure 6.** Wave train-induced pacemaker activity is a multicellular collective phenomenon. (**A**) Illuminated area (under constant light intensity) influences collective behaviour of excitable systems, transitioning between a stationary state for small illuminated areas and an oscillatory state for large illuminated areas. (**B**) In the oscillatory regime (large illuminated areas), illuminated cells in the centre (purple traces) are depolarised (both in silico and in vitro), while oscillatory behaviour still takes place in the bulk of the tissue (orange traces). This discards the simplest model of bi-stable limit cycles at the single-cell level.

illustrates that no oscillatory state could be induced with a high number of pulses for stimulus trains of high frequency (i.e. low stimulation period).

## Termination of collective pacemaker activity

A closer look at the high-frequency range showed that it not only prevented the initiation of ectopic pacemaker activity, but can also cause its termination (*Figure 5A*). In NRVM monolayers (3 biological replicates/monolayers), collective ectopic pacemaker activity was first initiated using a stimulus train of three pulses with a period of 500 ms. Once ongoing, eight pulses in the high-frequency range (5.000 Hz or period 200 ms) terminated the ectopic pacemaker activity and made the tissue go back to its resting state. When copying this experimental protocol in silico, qualitative correspondence was shown using four pulses with a period of 500 ms to initiate pacemaker activity, and seven pulses with a period of 180 ms to terminate it. Note that both experimentally and computationally, every pulse of the stimulus train is captured (orange traces), despite the fact that the voltage traces from points inside the illuminated region do not give that impression.

Once again, the qualitative in silico correspondence allows us to determine the specific conditions at which initiation and termination of collective ectopic pacemaker activity occur in more detail (*Figure 5B*). In the period versus number of pulses plot, which is a copy of the rightmost slice in *Figure 4C*, the region showing termination (red-lined) did not overlap with the region showing initiation (magenta-lined).

## The collective nature of wave train-induced pacemaker activity

After exploring collective bi-stability in practice, we aimed to gain a deeper understanding of this phenomenon. Bi-stability is a wide-spread phenomenon and can take place at the single-cell level (*Guevara, 2003*), which would be the simplest assumption for our monolayer system as well. When this would be the case, all cells would become oscillatory under illumination. We tested this hypothesis by making use of one of the four parameters we investigated earlier: the illuminated area. This parameter shows a transition from the stationary state towards the oscillatory state (*Figure 6A*) just like the light intensity parameter (*Figure 1B*) resulting in spontaneous oscillatory behaviour with waves emanating from the centre of the illuminated area for a large area size and at a constant low light

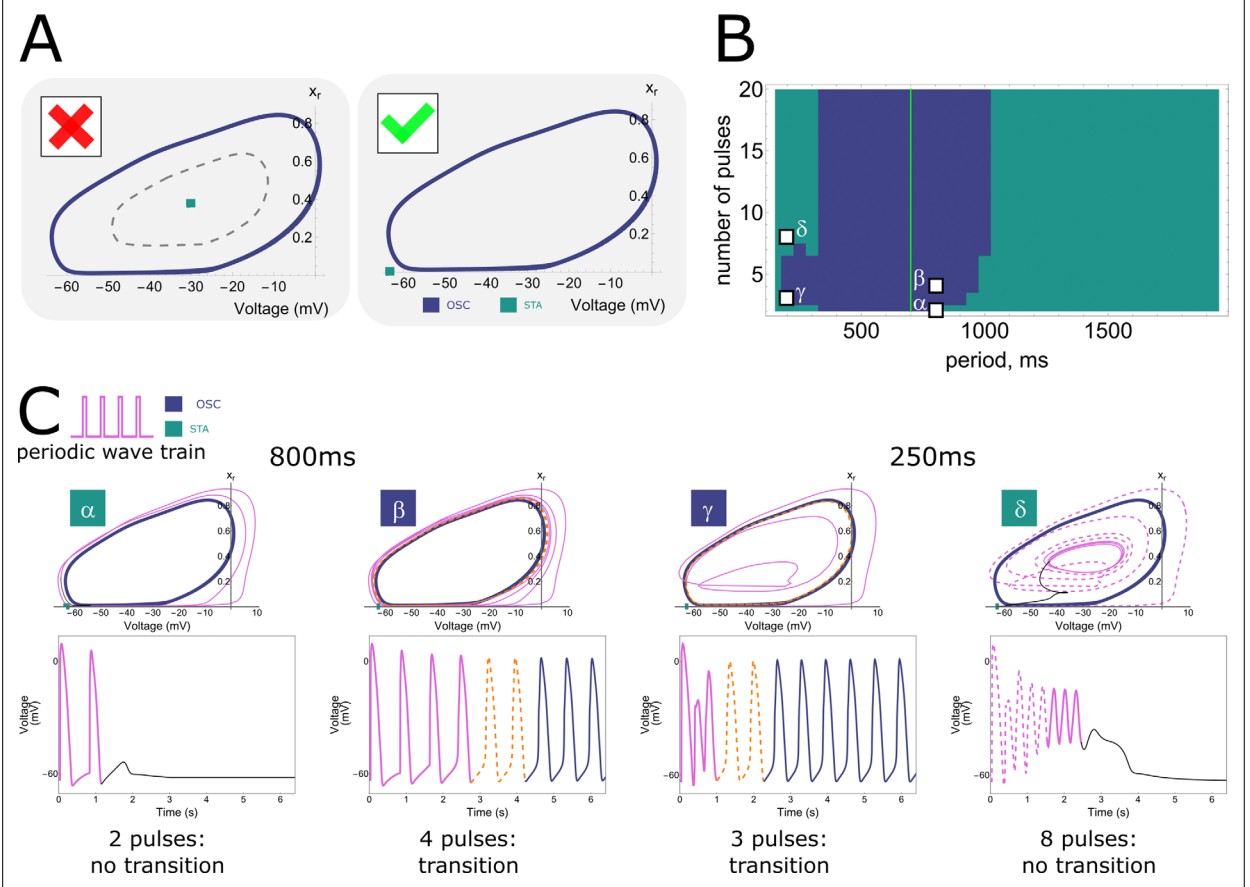

**Figure 7.** Phase plane projections of pulse-dependent collective state transitions. (**A**) Phase space trajectories (displayed in the voltage – $x_r$ plane) of the neonatal rat ventricular myocyte (NRVM) computational model show a limit cycle (OSC) that is not lying around a stable fixed point (STA). (**B**) Parameter space slice showing the relationship between stimulation period and number of pulses for a fixed illumination intensity (1.72 mW/mm$^2$) and size of the illuminated area (67 pixels edge length). Letters correspond to the graphs shown in (C). (**C**) Phase space trajectories for different combinations of stimulus train period and number of pulses ($\alpha$: 800 ms cycle length + 2 pulses, $\beta$: 800 ms cycle length + 4 pulses, $\gamma$: 250 ms cycle length + 3 pulses, $\delta$: 250 ms cycle length + 8 pulses). $\alpha$ and $\delta$ do not result in a transition from the resting state to ectopic pacemaker activity, as under these circumstances the system moves towards the stationary stable fixed point from outside and inside the stable limit cycle, respectively. However, for $\beta$ and $\gamma$, the stable limit cycle is approached from outside and inside, respectively, and ectopic pacemaker activity is induced.

The online version of this article includes the following figure supplement(s) for figure 7:

**Figure supplement 1.** Trajectory reconstruction using Takens time delay embedding.

intensity (*Figure 4D*). These circumstances allowed us to compare the voltage traces between the centre of the irradiated area and the bulk of the tissue, both in vitro and in silico. While the cells outside of the illuminated area showed oscillating membrane potentials, those in the centre of the illuminated region, that is those least affected by the coupling current of neighbouring cells, displayed a stable elevation of membrane potential. Hence, this implies that not all illuminated cells became oscillatory. We can therefore reject the simplest hypothesis of bi-stability at the single-cell level, but have to speak of a collective phenomenon. It is for this reason that throughout the manuscript the observed effects are referred to as 'collective' pacemaker activity.

## Insight into pulse-dependent collective state transitions

Knowing that we are de aling with a collective phenomenon, we wanted to gain a deeper understanding of the observed pulse-dependent collective transitions from stationary towards oscillatory pacemaker activity. In order to achieve this goal, in silico analysis was performed (*Figure 7*) because it offers the possibility to visualise all parameters relevant for the system. In every monolayer, we selected a single point from the centre of the illuminated region of the tissue and looked at its dynamic behaviour in phase space.

As was shown in all figures from *Figure 2* to *Figure 6*, illuminated cells do not show dampening oscillations when reaching the stationary state after a transition from the oscillatory state. This behaviour is justified when looking at a projection of the phase space of the model where the voltage $V$ and the gating variable $x_r$ for the rapid delayed outward rectifier K$^+$ current were chosen for the *x*- and *y*-axis, respectively (*Figure 7A*). The stable fixed point (STA) is not located inside the stable limit cycle (OSC) (right panel), hence showing no Hopf bifurcation (left panel) with its associated damped oscillations. The two attractor regions (STA and OSC) were also reconstructed from experimental data and showed two distinct trajectories (*Figure 7—figure supplement 1*).

Since we are dealing with a high-dimensional phase space in the in silico model, the visualisation of trajectories in time and the corresponding dynamic behaviour is rather complex. However, from the two-dimensional projections onto the $V$ and $x_r$ variables, we can see what causes the frequency and pulse dependency (*Figures 2–4*) of ectopic pacemaker activity initiation, as visualised in *Figure 7C*. Four different pairs of stimulation period and number of pulses were selected, corresponding to different behaviour areas in the parameter space slice for a fixed illumination intensity (1.72 mW/mm$^2$) and size of the illuminated area (67 pixels edge length) (*Figure 7B*). The first point ($\alpha$) comes from the region that shows no transition towards collective pacemaker activity for a low number of pulses (2) and intermediate stimulation periods of 800 ms. The phase space projection (*Figure 7C*) shows that the pulses cycle around the stable limit cycle and return back to the stable fixed point, indicating that two pulses are not enough to move within the attractor zone of the stable limit cycle. When the number of pulses is increased to 4 ($\beta$), the attractor zone of the stable limit cycle is reached from the outside, and collective ectopic pacemaker activity is induced. After switching to a faster stimulation period of 250 ms and sticking to just 3 pulses ($\gamma$), the phase space trajectories end up inside the stable limit cycle, but still within the attractor zone. Hence, also this time, collective pacemaker activity was induced. If we now increase the number of pulses again, this time to 8 ($\delta$), we are inside the stable limit cycle, but move again out of the region of attraction. Rather than producing oscillations, the system returns to the stationary state along dimensions other than those shown in *Figure 7C* (voltage and $x_r$), as evidenced by the phase space trajectory crossing itself. This return is mediated by the electrotonic current.

Thus, there is frequency selectivity due to the region of attraction of the stable limit cycle, where too low and too high frequencies result in stationary behaviour. Also, pulse accumulation is shown, where more pulses move the system closer to the attractor region. However, this is not a Hopf bifurcation, because in that case the system would not return to the stationary state when the number of pulses exceeds a critical threshold.

## Insight into the termination of ectopic pacemaker activity

Let us look once more at a cell inside the illuminated tissue undergoing a stimulus train with a short period of 180 ms from outside and extract all time-dependent variables. In response to a high number of stimuli (20 pulses), the time trace in the phase-space projection plot goes inside the limit cycle (*Figure 8A*) towards a quasi-steady state and returns back to the stationary state through a strangely curved line (black). This quasi-steady state is hidden within the oscillatory trajectory and no pacemaker activity is initiated. When the number of pulses is increased from 20 to 40, the voltage trace shows exactly the same behaviour, indicating that the number of pulses has no influence on the mechanism that moves the tissue back to the stationary state (*Figure 8B*). However, repolarisation reserve does have an influence, prolonging the transition when it is reduced (*Figure 8—figure supplement 1*). This effect can be observed either by moving further from the boundary of the illuminated region where the electrotonic influence from the non-illuminated region is weaker, or by introducing ionic changes, such as a reduction in $I_{Ks}$ and/or $I_{to}$. For example, because the electrotonic influence is weaker in the centre of the illuminated region, the voltage there is not pulled down towards the resting membrane potential as quickly as in cells at the border of the illuminated zone.

To understand what is going on, we studied single-cell dynamics under illumination. We already know from *Figure 6* that we are dealing with a purely collective phenomenon. Therefore, the single-cell dynamics itself showed nothing peculiar, that is a standard current–voltage relationship (*IV*) curve with a single stable equilibrium (not shown). To add a multicellular component to our single-cell model, we introduced a current that replicates the effect of cell coupling and its associated electrotonic influence. This so-called bias current equals the average in time and space of the current inside the

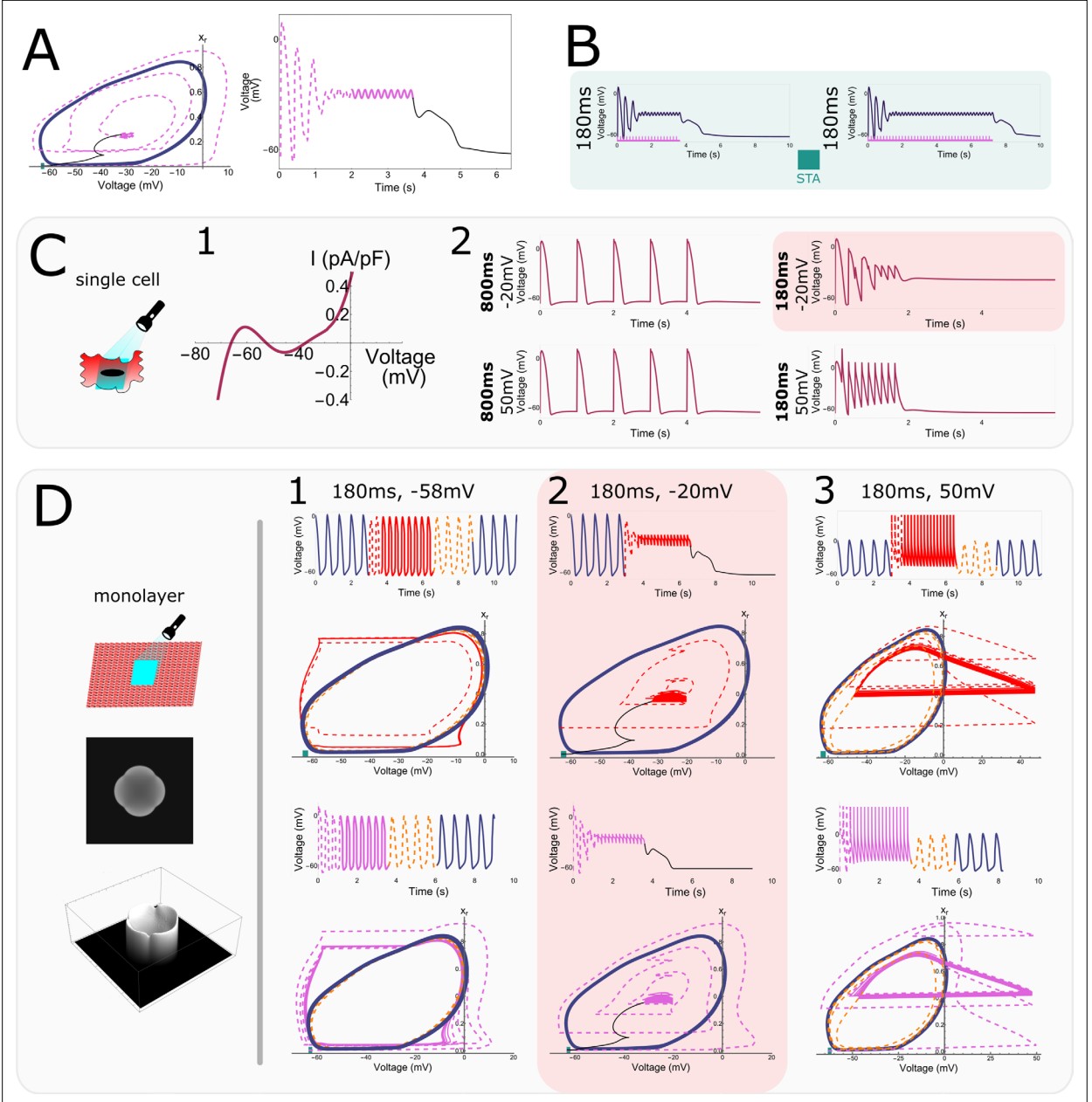

**Figure 8.** Single-cell 'hidden' bi-stability and its relation to high-frequency pacing. (**A**) Phase space trajectory and corresponding voltage trace at short pacing period (180 ms). (**B**) Formation of the quasi-steady state in the centre of an irradiated region during a train of 20 and 40 high-frequency (5.555 Hz or a 180-ms period) pulses. (**C**) Single-cell bi-stability. (1) Single-cell *IV* curve for an illuminated cell (1.72 mW/mm²) subject to a bias current. (2) Membrane voltage traces for stimulation periods of 1000 and 180 ms and stimulation amplitudes of −20 and 50 mV. A transition to the depolarised stable state occurred for short stimulation periods and low stimulation amplitudes. (**D**) Effect of spatially uniform voltage perturbations of different amplitudes (−58, −20, and 50 mV) and a period of 180 ms in cardiomyocyte monolayers. The top trace and phase space projection show the effect of perturbations when oscillation is ongoing, while the bottom ones show how perturbations affect the tissue in the stationary resting state. At an amplitude of −20 mV (2) oscillations were terminated when ongoing and could not be initiated when the tissue was in the resting state. In the other two cases (1,3), the opposite happened.

The online version of this article includes the following figure supplement(s) for figure 8:

**Figure supplement 1.** The effect of ionic changes on the termination of pacemaker activity.

illuminated region using wave trains with a period of 180 ms (the current produced during the quasi-steady state in *Figure 8B*). When this bias current is present, the *IV* curve of the single cell displays two stable equilibria and shows bi-stability (*Figure 8C-1*). The second stable equilibrium cannot be reached with a single pulse, but needs a stimulus train and can only be reached with short stimulation periods (180 *ms*) and intermediate stimulation amplitudes (–20 *mV*) (*Figure 8C-2*).

Building on this result, we next applied global stimuli with a period of 180 ms and amplitudes −58,–20, and 50 mV to a virtual cardiomyocyte monolayer with an illuminated square in the centre (*Figure 8D*). These pulses were applied when the tissue was either in the oscillatory state (top two rows) or the stationary state (bottom two rows). The stimulus amplitude of –20 mV that was necessary to get the single cell with a bias current into the second stable equilibrium, also terminates oscillatory behaviour (*Figure 8D-2*) while the other two amplitudes of –58 and 50 mV (*Figure 8D-1 and 3*) do not alter the behaviour of the tissue. However, when starting from the stationary state, the behaviour is opposite and the intermediate amplitude of –20 mV fails to induce collective oscillatory behaviour while the other two amplitudes do elicit this behaviour. We thus conclude that high stimulation frequencies and intermediate stimulus amplitudes are capable of terminating collective oscillatory behaviour by transitioning through a quasi-steady state.

We call this kind of bi-stability capable of terminating collective oscillatory behaviour '*hidden bi-stability*' for two reasons: (1) Single-cell bi-stability depends on electronic interactions. Hence, it can be said that this bi-stability is 'hidden' and only emerges in collective behaviour. (2) It is possible to escape the region of attraction of the oscillatory trajectory through opposite paths.

## It is complex, but not as complex as it seems

While it may be believed that the observed phenomena and explanations are intrinsic to entangled interactions specific to the complexity of a biological sample (in vitro) and a realistic computational model (in silico), we show here that this is not the case. By reducing the model equations, we show that collective oscillatory behaviour is not limited to the specific case of NRVMs. We started from the initial NRVM equations and reduced them to contain only a single depolarising current ($I_{Na}$), two repolarising currents ($I_{Kr}$ and the time-independent $I_{K1}$), and a spatially dependent optogenetic current ($I_{ChR2}$). Fast variables in these currents were assumed to keep their steady-state value, which caused them to be time-independent. The exact equations can be found in Appendix 1.

Using this simplified system of three variables (the voltage $V$, the closing gate of the Na$^+$ channel $h$, and the opening gate of the K$^+$ channel $x_r$), we were able to reproduce all key features (frequency selectivity, pulse accumulation, and bi-stability) of the more complex systems (*Figure 9*).

Just like in *Figure 7C*, the simplified model showed no initiation of collective pacemaker activity at a low number of two pulses and a low stimulation frequency of 0.444 Hz or a 2250-ms period, or at a high number of eight pulses and a high stimulation frequency of 1.666 Hz or a 600-ms period (*Figure 9A*). However, pacemaker activity was induced when applying three pulses and a low stimulation frequency or two pulses and a high stimulation frequency. At a low frequency, the phase space projection (again projected onto the $V$ and $x_r$ variable plane) shows that the limit cycle is approached from the outside, while at high frequencies it is approached from the inside. Altogether, these results show frequency selectivity and pulse accumulation in the simplified model (*Figure 9A*). To explain the findings in the rightmost panel of *Figure 9A*, we had a closer look at the single-cell dynamics under high stimulation frequencies. As shown in *Figure 9B*, four pulses at a low frequency (1.000 Hz or a 1000-ms period) resulted in normal behaviour, but high-frequency stimulation (2.000 Hz or a 500-ms period) pushed the cell towards a stably elevated membrane potential (*Figure 9B*). Looking at the influence of stimulus amplitude on the ability to get to this elevated membrane potential, it appears that after seven pulses, three of the four traces move towards the elevated membrane potential of –20 mV (*Figure 9C*). Panels B and C combined give us the same qualitative behaviour as seen in *Figure 8C*. The qualitative behaviours of *Figures 4A and 8D* were also reproduced and can be found in *Figure 9—figure supplement 1*.

## Discussion

In this study, in vitro observations were combined with in silico modelling to study and gain mechanistic insight into collective electrophysiological behaviour in NRVM monolayers.

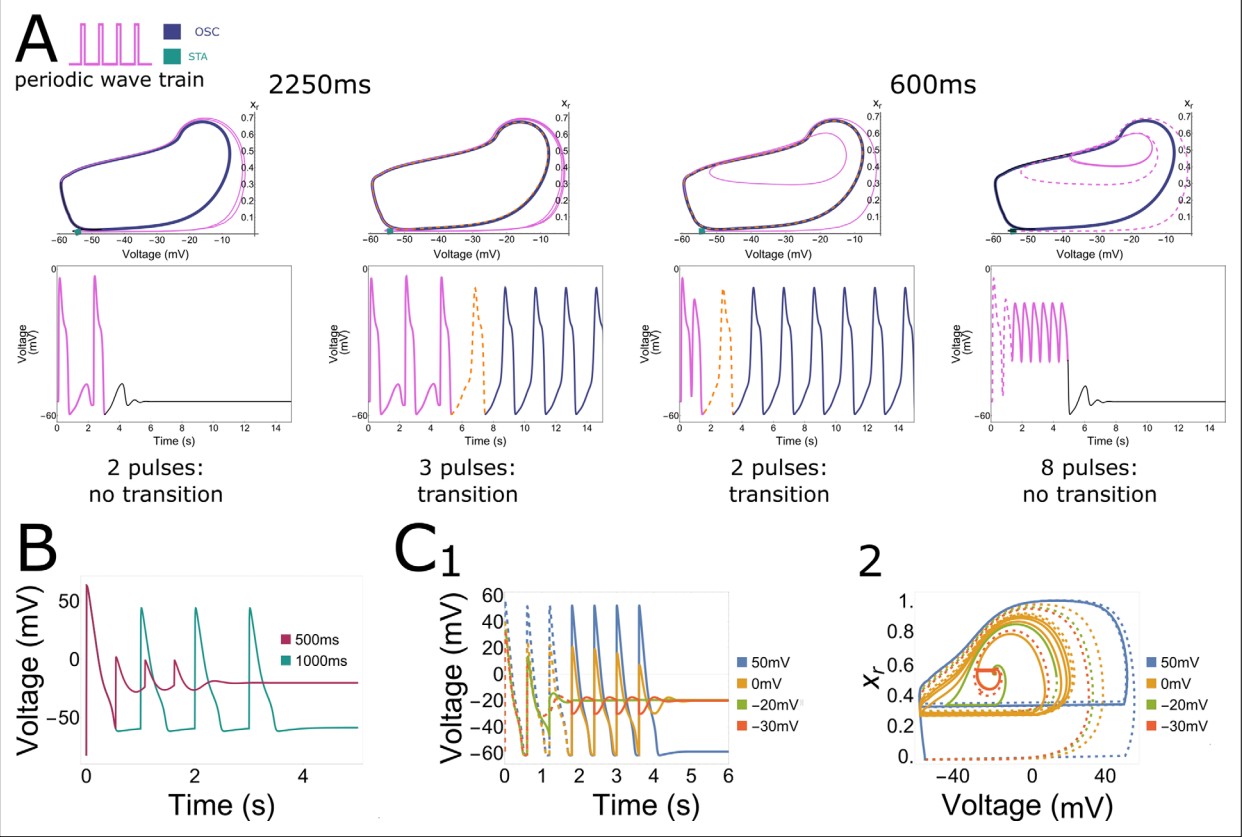

**Figure 9.** Collective pacemaker activity in a simplified in silico neonatal rat ventricular myocyte (NRVM) model. (**A**) Phase space trajectories of a point inside the illuminated region for different combinations of stimulus train period and number of pulses. A low stimulus frequency (0.444 Hz or a 2250-ms period) and a low number of two pulses, as well as a high stimulus frequency (1.666 Hz or a 600-ms period) with a high number of eight pulses both result in no transition from the resting state to ectopic pacemaker activity, moving towards the stationary stable fixed point from outside and inside the stable limit cycle, respectively. However, when applying three pulses with a low frequency (0.444 Hz or a 2250-ms period) or two pulses with a high-frequency (1.666 Hz or a 600-ms period), ectopic pacemaker activity is induced and the stable limit cycle is approached from outside and inside, respectively. (**B**) Under illumination, long period stimulation (1000 ms) shows a repolarised state after pacing, while short period stimulation shows a depolarised state (500 ms). (**C**) 1. Evolution of the voltage variable as a result of high-frequency voltage perturbations of high (blue trajectories) and low (green, orange, and red) magnitudes. The dashed parts of graphs demonstrate the transient non-periodic parts of the trajectories, the solid lines show the periodic and stationary states. 2. Corresponding trajectories in the $(V, x_r)$ phase plane.

The online version of this article includes the following figure supplement(s) for figure 9:

**Figure supplement 1.** Qualitative similarities between the simplified and complex reaction–diffusion model.

- Firstly, we demonstrated the existence of transitions from stationary cardiac monolayers towards ones that show pacemaker activity coming from an illuminated region, which are induced by wave trains of external pulses (*Figure 1*) as a manifestation of collective bi-stability. This transition between the two states is dependent on four variables: the light intensity applied to the illuminated region, the size of the illuminated region, the number of pulses of an incoming wave-train, and the frequency of the pulses of the incoming wave train (*Figures 2, 3, and 5*, *Figure 4—figure supplements 1 and 2*). In vitro experiments were qualitatively reproduced in silico, thereby establishing a link between model and experiment (*Figures 4 and 5* and *Figure 4—figure supplements 1 and 2*).
- Secondly, we studied the mechanisms underlying the initiation and termination of pacemaker activity. In the presence of a diffusion current (or bias current in the case of a single cell), the illuminated cells transition to a second equilibrium under high-frequency pacing. Because this second equilibrium only appears in a coupled system (or in the presence of a bias current), we can also speak of a hidden bi-stability. Hence, hidden bi-stability (high vs. low stationary resting membrane) is shown to underlie the collective phenomena (stationary vs. oscillatory behaviour) (*Figures 6–8*, *Figure 7—figure supplement 1*, *Figure 8—figure supplement 1*).

Below, we compare these phenomena to different types of arrhythmic events and mechanisms of bi-stability in cardiac tissue, and extend the relevance of our mechanisms to spiking behaviour in neuronal and other biophysical systems.

## Ectopic pacemaker activity in cardiology

The generation of ectopic beats is considered as one of the main mechanisms triggering and maintaining cardiac arrhythmias (*Pogwizd et al., 1992*). In most cases, ectopic sources are believed to be a result of triggered activity, which can manifest itself as early afterdepolarisations (EADs) or delayed afterdepolarisations (DADs) (*Qu et al., 2014*). Such ectopic sources are typically initiated by high-frequency pacing resulting in a $Ca^{2+}$ overload that triggers EADs and DADs (*Lerman et al., 2024*). However, as we did not observe $Ca^{2+}$ overload, the phenomenon observed in this study resembles induced pacemaker activity instead. In contrast with ectopy, this pacemaker activity is initiated at low pacing frequencies. Therefore, our findings might explain the phenomenon of bradycardia-induced triggered activity (*Brachmann et al., 1983*). The initiation of oscillations in our monolayers is the result of the spatial dynamics of the system, which stands in contrast with the frequency dependency of EADs that can be explained on a single-cell level (*Tran et al., 2009*; *Qu et al., 2014*). Our optogenetic current is absent in normal cardiac tissue, but its associated depolarisation could be caused by other late activating inward currents, which are also essential in the induction of EADs or DADs (*Song et al., 2008*; *Qu et al., 2014*): for example, the late Na current ($I_{Na,L}$, *Belardinelli et al., 2015*), reactivation of $I_{CaL}$ (*Tran et al., 2009*), or NCX activity (*Song et al., 2015*). In addition, collective bi-stability might be observed in the presence of oxidative stress, ischaemia, or chronic heart failure (*Belardinelli et al., 2015*; *Shryock et al., 2013*; *Undrovinas et al., 1999*), where there is depolarisation-induced automaticity (triggered activity) caused by a background current (e.g. $I_{Na,L}$).

The ectopic pacemaker activity in this study is a direct result of the combination of single-cell depolarisation with spatial effects. Due to the additional inward current produced by the activation of CheRiff, a single uncoupled illuminated cell goes to a steady depolarised state (*Figure 6B*). However, in a multicellular environment, there is a repolarising electrotonic coupling current working against this depolarising light-induced current. This repolarising current is a consequence of the interaction between depolarised and non-depolarised parts in the monolayer. By playing with the illumination intensity and the size of the illuminated region, we were able to balance these de- and re-polarising effects to create a bi-stable regime (STA vs. OSC). This also explains why there are no oscillations present in the centre of a large illuminated region, since there the electrotonic current is lower than at its edges (*Figure 6C*). On the other hand, if the electrotonic current is too strong, it will prevent depolarisation by the optogenetic current and, as a consequence, the system will stay in the stationary state. The balancing of currents also explains why the oscillatory boundary moves closer to the edge of the illuminated region at higher light intensities, since under those circumstances the depolarising current will overpower the spatial effects. However, when the illumination is very strong, it is possible to have oscillations that do not need to be induced (*Figure 1B*) and can even come from the corners of the illuminated region (*Figure 1—figure supplement 3*, *Teplenin et al., 2018*). This last-mentioned phenomenon appears because the curvature of the boundary between illuminated and not-illuminated regions influences the electrotonic current.

## Interplay between microscopic (single cell) and macroscopic (monolayer) behaviour

Collective bi-stability, as described in the previous section, is a bi-stability between two distinct behaviours (stationary STA and oscillatory OSC behaviour) that occur at the macroscopic level (monolayer). However, by looking more deeply at the dynamics that happen at the microscopic level (single cell), we can get more insight into the underlying mechanism(s). At the single-cell level, the dynamic transition between a stationary monolayer and one showing induced pacemaker activity relies on two main processes:

1. For the initiation of pacemaker activity, pulse accumulation (at intermediate frequencies) is necessary to bring the system into the attracting region of the oscillatory state.
2. For the termination of pacemaker activity, an interaction takes place between 'hidden' bi-stable dynamics and coupling currents at high frequencies.

Both these processes occur as a consequence of wave pulses outside of the illuminated region. At low frequencies, no oscillatory activity can be induced. At intermediate frequencies, it is possible to escape the attracting zone of the stationary state and cross the separatrix into the attracting zone of the oscillatory state. However, for even higher frequencies, the system gets pushed out of the attracting zone of the oscillatory state again and through hidden bi-stability, it returns to the stationary state.

Our data on the termination mechanism of induced pacemaker activity (*Figure 8*) indicate that we are not dealing with classical STA–OSC bi-stability (*Winfree, 2001*). In systems displaying classical bi-stability between a limit cycle and the resting state, pacemaker annihilation can be achieved by applying sufficiently high-frequency voltage perturbations. Through pulse accumulation, it is then possible to exit the basin of attraction in the reverse direction from which it was entered. In contrast, our system enters (from outside the limit cycle) and leaves (towards the inside of the limit cycle) the attraction zone in opposite directions. This was concluded from the absence of a Hopf bifurcation in our system (*Figure 7*).

The seemingly fragile nature of hidden bi-stability might be a requirement for coexistence of excitable and bi-stable regimes. Otherwise, if the hills and dips of the N-shaped steady-state *IV* curve were large (*Figure 8C-1*), they would have similar magnitudes as the large currents of fast ion channels, preventing the subtle interaction between these strong ionic cell currents and the small repolarising coupling currents (≈0.1 pA).

Although the termination mechanism was shown to be amplitude-dependent (*Figure 8C, D*), the appropriate perturbation amplitudes are 'automatically' selected by a high-frequency train of travelling waves of excitation without any a priori knowledge about the state of the system. This wave train terminated ectopic triggers (*Figure 2*), which is in accordance with previous publications (*Moak and Rosen, 1984*; *Glikson et al., 2021*). These publications also showed that uniform high voltage shocks are unable to terminate ectopic sustained activity, similar to our findings (*Figure 8*).

Although its manifestations are clear in a monolayer system, hidden bi-stability might be challenging to observe in single-cell patch-clamp experiments. In *Figure 8C-1*, the height of dips and hills of the N-shaped steady-state *IV* curve under bias current are ≈0.2 pA/pF. While it is possible to perform patch-clamping on individual cells in a monolayer experiencing diffusion currents, measurements of small amplitudes can be hindered by the statistical and instrumental errors of voltage-clamp recordings. The voltage bi-stability we are looking for can, for example, easily be lost via a shift in current balance due to the non-zero resistance of the patch pipette in current clamp mode. This might explain why the phenomenon of hidden bi-stability has only rarely been reported, for example, in an in silico study dedicated to myotony in muscle fibres (*Cannon et al., 1993*).

## Extrapolation and translation

All key aspects of the initiation and termination of induced pacemaker activity, that is the points itemised above were also demonstrated in a simplified 3-variable reaction–diffusion model (*Figure 9*, *Figure 9—figure supplement 1*), implicating the more general nature of this observed phenomenon. A rigorous mathematical analysis can be accomplished in future work, but might be challenging due to the lack of efficient analytical tools to describe fully developed limit cycle oscillations in reaction–diffusion systems (*Sherratt and Smith, 2008*). Based on the reproduction of our results in such a simplified model, we showed that we do not need the full complexity of an ionic cardiomyocyte model, enabling extrapolation and translation of our results towards other scientific disciplines.

Frequency-dependent initiation and termination of oscillatory activity is not exclusive to cardiac systems, but is well known in a variety of non-linear biophysical systems. For example, the actomyosin cortex in the social amoeba *Dictyostelium discoideum* responds to short pulses of 3',5'-cyclic adenosine monophosphate (cAMP) by damped oscillations in average actin content, thereby allowing fine time-selective responses to either internal (pseudopodia initiation) or external (cAMP release) perturbations (*Westendorf et al., 2013*). Moreover, in synthetic gene regulatory networks, subthreshold stimuli might be used to discriminate environmental stimuli (e.g. growth factor release, heat shock) (*Guantes and Poyatos, 2006*). Also, neurons can differentiate external stimuli depending on their frequency. As mentioned in the introduction, the frequency-dependent initiation and termination of oscillatory activity is known as '*resonance*' in neuroscience. It is interesting to compare the resonance properties of our system with those actively studied in neurons.

Resonant neurons, also known as resonate-and-fire neurons or type II excitability neurons (*Izhikevich, 2000*), are governed by sub-threshold oscillations initiated due to proximity of the resting state to a Hopf bifurcation (*Hutcheon and Yarom, 2000*; *Roach et al., 2018*). They react to a stimulus train as follows: The first pulse initiates a damped sub-threshold oscillation of the membrane potential, while the effect of the second pulse depends on its timing. If timed correctly, it adds to the first pulse and either increases the amplitude of the sub-threshold oscillation or evokes repetitive spiking (transition to a limit cycle). Thus, the response and spiking of the resonator neuron depends on the frequency content of the input. These types of spiking neurons are usually associated with Bogdanov–Takens bifurcations (*Izhikevich, 2000*). However, another type of bifurcation, a saddle-node-loop (SNL) bifurcation, is also capable of generating spiking neurons and allows a bi-stability regime, in which the stable state lies outside the limit cycle similar to our case (*Figure 7*).

SNL bifurcations were previously identified in neurons (*Hesse et al., 2017*; *Hesse and Schreiber, 2019*; *Schleimer et al., 2021*), in a model of transcriptional regulation of the cell (*Ciliberto et al., 2005*), and in the design of combined genetic switches (*Perez-Carrasco et al., 2018*). Our result might represent the emergent analogue of an SNL bifurcation with resonant transitions on a neuronal/non-linear network level. SNL bifurcations have already been identified in the networks of coupled Kuramoto limit cycle oscillators both in a broad (*Martens et al., 2009*; *Childs and Strogatz, 2008*) and in a neuroscience context (*Buendía et al., 2021*). In these studies, SNL bifurcations were identified using dimensionality reduction techniques while ignoring the whole wealth of transient and finite-size phenomena. Moreover, the Kuramoto model is a model without many physical details, describing a weak forcing regime on an idealised limit cycle oscillator, in which oscillations cannot be destroyed but only phase shifted. In our case, we provide a more realistic biophysically motivated model, which allows a strong forcing regime and destruction of oscillations due to the phenomenon of hidden bi-stability. This gives us reason to believe that idealised Kuramoto models could possess resonant properties if augmented/modified with sufficient physical/biophysical details. SNL bifurcations and hidden bi-stability could also be studied in a range of other unexplored biophysical situations including high-frequency stimulation for the termination of neuronal oscillations during Parkinson's disease (*Touboul et al., 2020*), multi-compartment cable computational models of the neuron (*Schwemmer and Lewis, 2012*), and bi-stability in neurons in vitro (*Le et al., 2006*). Additional studies can be carried out in neuroscience by local optogenetic activation of depolarising ion channels, both in vivo (*Lu et al., 2015*) and in silico in non-local models (*Selvaraj et al., 2015*; *Heitmann et al., 2017*).

## Conclusion

Using both in vitro and in silico models, we explored a mechanism for frequency-dependent initiation and termination of oscillatory activity (i.e. induced pacemaker activity) in non-homogeneous cardiac tissue. This mechanism could be extended to more general excitable systems and is in neuroscience also known under the term resonance. Our collective 'resonance' mechanism could be used to study or control collective states in cardiac but also other excitable systems.

## Materials and methods

### Preparation of monolayers of NRVMs expressing the light-activatable cation channel CheRiff

All animal experiments were reviewed and approved by the Animal Experiments Committee of the Leiden University Medical Center and carried out in accordance with Directive 2010/63/EU of the European Union on the protection of animals used for scientific purposes. Monolayers of NRVMs were established as previously described by *Engels et al., 2015*. Briefly, the hearts were excised from anaesthetised 2-day-old Wistar rats. The ventricles were cut into small pieces and dissociated in a solution containing 450 U/ml collagenase type I (Worthington, Lakewood, NJ) and 18.75 Kunitz/ml DNase I (Sigma-Aldrich, St. Louis, MO). The resulting cell suspension was enriched for cardiomyocytes by preplating for 120 min in a humidified incubator at 37°C and 5% $CO_2$ using Primaria culture dishes (Becton Dickinson, Breda, the Netherlands). Finally, the cells were seeded on round glass coverslips ($d = 15$ mm; Gerhard Menzel, Braunschweig, Germany) coated with bovine fibronectin (100 µg/ml; Sigma-Aldrich) to establish confluent monolayers. After incubation overnight in an atmosphere of humidified 95% air–5% $CO_2$ at 37°C, these monolayers were treated with Mitomycin-C

(10 μg/ml; Sigma-Aldrich) for 2 hr to minimise proliferation of the non-cardiomyocytes. At day 4 of culture, the NRVM monolayers were incubated for 20–24 hr with lentiviral vector particles encoding CheRiff (*Hochbaum et al., 2014*), a light-gated depolarising ion channel of the channelrhodopsin family (*Wang et al., 2015*), at a dose resulting in homogeneous transduction of nearly 100% of the cells (*Feola et al., 2017*; *Ördög et al., 2023*). Next, the cultures were washed once with phosphate-buffered saline, given fresh culture medium and kept under culture conditions for 3–4 additional days.

## Optical voltage mapping and patterned illumination of NRVM monolayers

After 8–10 days of culturing, NRVM monolayers were optically mapped using the voltage-sensitive dye di-4-ANBDQBS (52.5 μM final concentration; ITK diagnostics, Uithoorn, the Netherlands) as reported previously (*Feola et al., 2017*). The mapping setup (*Figure 1—figure supplement 1*) was based on a 100 × 100 pixel CMOS Ultima-L camera (Scimedia, Costa Mesa, CA). The field of view was 16 × 16 mm resulting in a spatial resolution of 160 μm/pixel. For targeted illumination of mono-layers, the setup was optically conjugated to a digitally controlled micro-mirror device, the Polygon 400 (Mightex Systems, Toronto, ON), with a high power blue (470 nm) LED (BLS-LCS-0470-50-22-H, Mightex Systems). Before starting the actual experiments, all monolayers were mapped during 1 Hz electrical point stimulation to check baseline conditions. Electrical stimulation was performed by applying 10-ms-long rectangular electrical pulses with an amplitude of 8 V to a bipolar platinum electrode with a spacing of 1.5 mm between anode and cathode. Only cultures with an action potential duration (APD) at 80% repolarisation (APD80) below 350 ms and a conduction velocity (CV) above 18 cm/s were used for further experiments. Stimuli were applied in trains that varied in number and period. Whenever the outcome was reproducible three consecutive times in a single monolayer, it was counted as successful and included as one measurement in the total dataset. The constant light intensity was varied in the range of (0.03125–0.25 mW/mm$^2$) to search the critical value for the bi-stable regime. For experiments in which the light intensity was varied the rectangular-shaped area of illumination was in the range of 4 × 4 to 6 × 6 mm. The highest achievable irradiation intensity (0.3125 mW/mm$^2$) was used to perform size modulation experiments. The resulting electrical activity was recorded for 6–24 s at exposure times of 6 ms per frame.

## Post-processing of optical mapping data

Data processing was performed using specialised BV Ana software (Scimedia), ImageJ (*Schneider et al., 2012*) and custom-written scripts in Wolfram Mathematica (Wolfram Research, Hanborough, Oxfordshire, United Kingdom). APD and CV were calculated as described previously (*Feola et al., 2017*). To prepare representative frames of wave propagation, optical mapping videos were filtered with a spatial averaging filter (3 × 3 stencil) and a derivative filter.

## Attractor reconstruction

Multiple attractor reconstruction is based on Takens's embedding theorem (*Takens, 1981*) by plotting phase space trajectories in $(V(t), V(t + \tau))$ coordinates. The optimal time delay $\tau$ was determined by calculating the minimum of the autocorrelation function (*Clemson and Stefanovska, 2014*). Linear drift removal was applied before plotting phase space trajectories.

## Detailed electrophysiological model

In this study, the Majumder–Korhonen model (*Majumder et al., 2016*) of NRVM monolayers was used, whose monodomain reaction–diffusion equation is formulated as follows:

$$\frac{\partial V}{\partial t} = \nabla \cdot (D \nabla V) - \frac{I_{ion} + I_{ChR2}(x, y)}{C_m}, \tag{1}$$

where $I_{ion}$ is the sum of all ionic currents, $\mathcal{D}$ is the diffusion coefficient equal to 0.00095 cm$^2$/ms and $I_{ChR2}(x, y)$ is the spatially controlled optogenetic current. In comparison to the original model, $I_{Na}$ was increased 1.3-fold and, $I_{CaL}$ and $I_{Kr}$ were both reduced 0.7-fold. The steady-state voltage dependency for the inactivation variable $h$ of the fast Na$^+$ current was changed to $h_\infty = [1 + e^{\frac{(72+V)}{6.07}}]^{-1}$, where $V$ is the transmembrane potential. The resulting CV was 20 cm/s. The optogenetic current $I_{ChR2}$ was formulated based on the *Boyle et al., 2013* $I_{ChR2}$ model with the introduction of rectifying conduction

properties and voltage-dependent state transitions (*Williams et al., 2013*). The irradiation intensity varied in the range of 0.165–0.175 mW/mm$^2$.

The forward Euler method was used to integrate the equations with a time step $\Delta t = 0.02 \, \text{ms}$ and a centred finite-differencing scheme to discretise the Laplacian with a space step of $\Delta x = 0.0625 \, \text{mm}$. The total computational domain size was 256 × 256 grid points, while the centrally irradiated square with the optogenetic current consisted of 80 × 80 grid points. To create stationary initial conditions, the model was integrated for 2 min before a train of stimuli was applied from the upper border of the domain.

## Acknowledgements

We would like to thank Leon Glass and Desmond Kabus for useful discussions that improved the manuscript and Annemarie Kip and Cindy Bart for the production of the CheRiff-encoding lentiviral vector LV.GgTnnt2.CheRiff eYFP.WHVoPRE. This work was supported by the European Research Council (ERC consolidator grant 101044831 to DAP) and the Netherlands Organization for Scientific Research (NWO Vidi grant 917143 to DAP and ZonMW Off Road grant 04510012110051 to TDC).

## Additional information

### Funding

| Funder | Grant reference number | Author |
| --- | --- | --- |
| European Research Council | 101044831 | Daniël A Pijnappels |
| Nederlandse Organisatie voor Wetenschappelijk Onderzoek | 917143 | Daniël A Pijnappels |
| ZonMw | 04510012110051 | Tim De Coster |

The funders had no role in study design, data collection, and interpretation, or the decision to submit the work for publication.

### Author contributions

Alexander S Teplenin, Conceptualization, Formal analysis, Investigation, Methodology, Software, Visualization, Writing – review and editing; Nina N Kudryashova, Rupamanjari Majumder, Software; Antoine AF de Vries, Resources, Writing – review and editing; Alexander V Panfilov, Conceptualization, Writing – review and editing; Daniël A Pijnappels, Conceptualization, Funding acquisition, Project administration, Resources, Supervision, Writing – review and editing; Tim De Coster, Conceptualization, Formal analysis, Funding acquisition, Supervision, Visualization, Writing – original draft, Writing – review and editing

### Author ORCIDs

Alexander S Teplenin ⓘ https://orcid.org/0000-0001-7841-376X
Daniël A Pijnappels ⓘ https://orcid.org/0000-0001-6731-4125
Tim De Coster ⓘ https://orcid.org/0000-0002-4942-9866

### Ethics

All animal experiments were reviewed and approved by the Animal Experiments Committee of the Leiden University Medical Center and carried out in accordance with Directive 2010/63/EU of the European Union on the protection of animals used for scientific purposes.

Reviewer #1 (Public review): https://doi.org/10.7554/eLife.107072.3.sa1
Reviewer #2 (Public review): https://doi.org/10.7554/eLife.107072.3.sa2
Author response https://doi.org/10.7554/eLife.107072.3.sa3

## Additional files

### Supplementary files
MDAR checklist

### Data availability
The raw mapping data supporting the findings of this study are available from Dryad (https://doi.org/10.5061/dryad.dz08kpsbh) and can be converted to numpy arrays using the open source software sappho (*Kabus et al., 2024*), which is available from Gitlab (https://gitlab.com/heartkor/sappho; *Kabus, 2025*). All simulation results can be reproduced using the code that is made available in Zenodo (https://doi.org/10.5281/zenodo.17859360).

The following datasets were generated:

| Author(s) | Year | Dataset title | Dataset URL | Database and Identifier |
| --- | --- | --- | --- | --- |
| Teplenin AS, Kudryashova NN, Majumder R, de Vries AA, Panfilov AV, Pijnappels DA, De Coster T | 2025 | Data from: Atypical collective oscillatory activity in cardiac tissue uncovered by optogenetics | https://doi.org/10.5061/dryad.dz08kpsbh | Dryad Digital Repository, 10.5061/dryad.dz08kpsbh |
| Teplenin AS, Kudryashova NN, Majumder R, de Vries AA, Panfilov AV, Pijnappels DA, De Coster T | 2025 | Code from: Atypical collective oscillatory activity in cardiac tissue uncovered by optogenetics | https://doi.org/10.5281/zenodo.17859360 | Zenodo, 10.5281/zenodo.17859360 |

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

## Appendix 1

### Supplementary methods

Simplified reaction–diffusion model

The simplified reaction–diffusion model consists of one voltage variable, two gating variables for the depolarising and hyperpolarising currents, and an inhomogeneity parameter $C(x, y)$ :

$$\frac{\partial V}{\partial t} = D\Delta V - 6(1 + e^{-\frac{(45 + V)}{6.5}})^{-3}j^2(V - 62.84) \tag{2}$$

$$-(1 + e^{\frac{V + 9}{22.4}})^{-1}x_r 0.011536(V + 86.8358) \tag{3}$$

$$-0.6I_{K1} - 0.24(10.6408 - 14.6408e^{\frac{-V}{42.7671}})C(x, y) \tag{4}$$

$$\frac{\partial j}{\partial t} = \frac{(j_\infty(V) - j)}{\tau_j(V)} \tag{5}$$

$$\frac{\partial x_r}{\partial t} = \frac{(x_{r_\infty}(V) - x_r)}{\tau_j(V)} \tag{6}$$

where $D$ is the diffusion coefficient equal to 0.0003 cm²/ms. The $I_{K1}$ current formulation was copied from *Majumder et al., 2016*. The $C(x, y)$ was set to 0.1518115 inside the illuminated square and to 0 outside this. $j_\infty(V), x_{r_\infty}(V), \tau_j(V), \tau_j(V)$ are described as:

$$j_\infty(V) = (1 + e^{\frac{(68 + V)}{6.07}})^{-1} \tag{7}$$

$$x_{r_\infty}(V) = (1 + e^{\frac{-(21.5 + V)}{7.5}})^{-1} \tag{8}$$

$$\tau_x(V) = \begin{cases} 2\left(\frac{0.00061(V + 38.9)}{e^{0.145(V+38.9)} - 1} + \frac{0.00138(V + 14.2)}{1 - e^{-0.123(V+14.2)}}\right)^{-1}, & V > -57 \\ 4\left(\frac{0.00061(V + 38.9)}{e^{0.145(V+38.9)} - 1} + \frac{0.00138(V + 14.2)}{1 - e^{-0.123(V+14.2)}}\right)^{-1}, & V \leq -57 \end{cases} \tag{9}$$

$$\tau_j(V) = \begin{cases} 11.63\left(1 + e^{-0.1(V+32)}\right)e^{2.535*10^{-7}V}, V \geq -40 \\ 3.49\left(\frac{-127140(V + 37.78)}{1 + e^{0.311(V+79.23)}}e^{0.2444V} - \frac{3.474}{10^5 e^{0.04391*V}} + \frac{0.1212e^{-0.01052*V}}{1 + e^{0.1378(V+40.14)}}\right)^{-1}, & V < -40 \end{cases} \tag{10}$$

The forward Euler method was used to integrate the equations with a time step $\Delta t = 0.01$ ms and a centred finite-differencing scheme to discretise the Laplacian with a space step of $\Delta x = 0.0625$ mm. The total computational domain size was 256 × 256 grid points while the square with the 'background' current consisted of 80 × 80 grid points. To create stationary initial conditions, the model was integrated for 2 min before a train of stimuli was applied from the upper border of the domain.

