## [Editor Report · eLife Assessment]

This **important** work provides mechanistic insights into the development of cardiac arrhythmia and establishes a new experimental use case for optogenetics in studying cardiac electrophysiology. The agreement between computational models and experimental observations provides a **convincing** level of evidence that wave train-induced pacemaker activity can originate in continuously depolarized tissue, with the limitation that there may be differences between depolarization arising from constant optogenetic stimulation, as opposed to pathophysiological tissue depolarization. Future experiments in vivo and in other tissue preparations would extend the generality of these findings.

---

## [Referee Report · Reviewer #1 (Public review)]

Summary:

The study by Teplenin and coworkers assesses the combined effects of localized depolarization and excitatory electrical stimulation in myocardial monolayers. They study the electrophysiological behaviour of cultured neonatal rat ventricular cardiomyocytes expressing the light-gated cation channel Cheriff, allowing them to induce local depolarization of varying area and amplitude, the latter titrated by the applied light intensity. In addition, they used computational modeling to screen for critical parameters determining state transitions, and for dissecting the underlying mechanisms. Two stable states, thus bistability, could be induced upon local depolarization and electrical stimulation, one state characterized by a constant membrane voltage and a second spontaneously firing, thus oscillatory state. The resulting 'state' of the monolayer was dependent on the duration and frequency of electrical stimuli, as well as the size of the illuminated area and the applied light intensity determining the degree of depolarization as well as the steepness of the local voltage gradient. In addition to the induction of oscillatory behaviour, they also tested frequency-dependent termination of induced oscillations.

Strengths:

The data from optogenetic experiments and computational modelling provide quantitative insights into the parameter space determining the induction of spontaneous excitation in the monolayer. The most important findings can also be reproduced using a strongly reduced computational model, suggesting that the observed phenomena might be more generally applicable.

Weaknesses:

While the study is thoroughly performed and provides interesting mechanistic insights into scenarios of ventricular arrhythmogenesis in the presence of localized depolarized tissue areas, the translational perspective of the study remains relatively vague. In addition, the chosen theoretical approach and the way the data is presented might make it difficult for the wider community of cardiac researchers to understand the significance of the study.

Comments on Revision:

The provided revisions address some of the raised concerns, but they do not change my general assessment of the paper, including its strengths and weaknesses.

---

## [Referee Report · Reviewer #2 (Public review)]

In the presented manuscript, Teplenin and colleagues use both electrical pacing and optogenetic stimulation to create a reproducible, controllable source of ectopy in cardiomyocyte monolayers. To accomplish this, they use a careful calibration of electrical pacing characteristics (i.e., frequency, number of pulses) and illumination characteristics (i.e., light intensity, surface area) to show that there exists a "sweet spot" where oscillatory excitations can emerge proximal to the optogenetically depolarized region following electrical pacing cessation, akin to pacemaker cells. Furthermore, the authors demonstrate that a high-frequency electrical wave-train can be used to terminate these oscillatory excitations. The authors observed this oscillatory phenomenon both in vitro (using neonatal rat ventricular cardiomyocyte monolayers) and in silico (using a computational action potential model of the same cell type). These are surprising findings and provide a novel approach for studying triggered activity in cardiac tissue.

The study is extremely thorough and one of the more memorable and grounded applications of cardiac optogenetics in the past decade. One of the benefits of the authors' "two-prong" approach of experimental preps and computational models is that they could probe the number of potential variable combinations much deeper than through in vitro experiments alone. The strong similarities between the real-life and computational findings suggest that these oscillatory excitations are consistent, reproducible, and controllable.

Triggered activity, which can lead to ventricular arrhythmias and cardiac sudden death, has been largely contributed to sub-cellular phenomena, such as early or delayed afterdepolarizations, and thus to date has largely been studied in isolated single cardiomyocytes. However, these findings have been difficult to translate to tissue- and organ-scale experiments, as well-coupled cardiac tissue has notably different electrical properties. This underscores the significance of the study's methodological advances: use of a constant depolarizing current in a subset of (illuminated) cells to reliably result in triggered activity could facilitate the more consistent evaluation of triggered activity at various scales. An experimental prep that is both repeatable and controllable (i.e., both initiated and terminated through the same means) is a boon for further inquiry.

The authors also substantially explored phase space and single cell analyses to document how this "hidden" bi-stable phenomenon can be uncovered during emergent collective tissue behavior. Calibration and testing of different aspects (e.g.: light intensity, illuminated surface area, electrical pulse frequency, electrical pulse count) and other deeper analyses, as illustrated in Figures S3-S8 and Video S1, are significant and commendable.

Given the study is computational, it is surprising that the authors did not replicate their findings using well-validated adult ventricular cardiomyocyte action potential models, such ten Tusscher 2006 or O'Hara 2011. This may have felt out-of-scope, given the nice alignment of rat cardiomyocyte data between in vitro and in silico experiments. However, it would have been helpful peace-of-mind validation, given the significant ionic current differences between neonatal rat and adult ventricular tissue. It is not fully clear whether the pulse trains could have resulted in the same bi-stable oscillatory behavior, given the longer APD of humans relative to rats. The observed phenomenon certainly would be frequency-dependent and would have required tedious calibration for a new cell type, albeit partially mitigated by the relative ease of in silico experiments.

There are likely also mechanistic differences between this optogenetically-tied oscillatory behavior and triggered activity observed in other studies. This is because the constant light-elicited depolarizing current is disrupting the typical resting cardiomyocyte state, thereby altering the balance between depolarizing ionic currents (such as Na+ and Ca2+) and repolarizing ionic currents (such as K+ and Ca2+). The oscillatory excitations appear to later emerge at the border of the illuminated region and non-stimulated surrounding tissue, which is likely an area of high source-sink mismatch. The authors appear to acknowledge differences in this oscillatory behavior and previous sub-cellular triggered activity research in their discussion of ectopic pacemaker activity, which are canonically observed in genetic, pharmacologic, or pathological ionic conditions. Regardless, it is exciting to see new ground being broken in this difficult-to-characterize experimental space, even if the method illustrated here may not necessarily be broadly applicable.

Comments on revisions:

I have read the authors' rebuttal to our earlier comments and do not have any further questions or comments. Thank you for implementing the minor improvements to Figure visualizations and for creating Video S1 to accompany the article.

---

## [Author Response]

The following is the authors’ response to the original reviews

**Public Reviews:**

**Reviewer #1 (Public review):**
Summary:The study by Teplenin and coworkers assesses the combined effects of localized depolarization and excitatory electrical stimulation in myocardial monolayers. They study the electrophysiological behaviour of cultured neonatal rat ventricular cardiomyocytes expressing the light-gated cation channel Cheriff, allowing them to induce local depolarization of varying area and amplitude, the latter titrated by the applied light intensity. In addition, they used computational modeling to screen for critical parameters determining state transitions and to dissect the underlying mechanisms. Two stable states, thus bistability, could be induced upon local depolarization and electrical stimulation, one state characterized by a constant membrane voltage and a second, spontaneously firing, thus oscillatory state. The resulting 'state' of the monolayer was dependent on the duration and frequency of electrical stimuli, as well as the size of the illuminated area and the applied light intensity, determining the degree of depolarization as well as the steepness of the local voltage gradient. In addition to the induction of oscillatory behaviour, they also tested frequency-dependent termination of induced oscillations.Strengths:The data from optogenetic experiments and computational modelling provide quantitative insights into the parameter space determining the induction of spontaneous excitation in the monolayer. The most important findings can also be reproduced using a strongly reduced computational model, suggesting that the observed phenomena might be more generally applicable.Weaknesses:While the study is thoroughly performed and provides interesting mechanistic insights into scenarios of ventricular arrhythmogenesis in the presence of localized depolarized tissue areas, the translational perspective of the study remains relatively vague. In addition, the chosen theoretical approach and the way the data are presented might make it difficult for the wider community of cardiac researchers to understand the significance of the study.
**Reviewer #2 (Public review):**
In the presented manuscript, Teplenin and colleagues use both electrical pacing and optogenetic stimulation to create a reproducible, controllable source of ectopy in cardiomyocyte monolayers. To accomplish this, they use a careful calibration of electrical pacing characteristics (i.e., frequency, number of pulses) and illumination characteristics (i.e., light intensity, surface area) to show that there exists a "sweet spot" where oscillatory excitations can emerge proximal to the optogenetically depolarized region following electrical pacing cessation, akin to pacemaker cells. Furthermore, the authors demonstrate that a high-frequency electrical wave-train can be used to terminate these oscillatory excitations. The authors observed this oscillatory phenomenon both in vitro (using neonatal rat ventricular cardiomyocyte monolayers) and in silico (using a computational action potential model of the same cell type). These are surprising findings and provide a novel approach for studying triggered activity in cardiac tissue.The study is extremely thorough and one of the more memorable and grounded applications of cardiac optogenetics in the past decade. One of the benefits of the authors' "two-prong" approach of experimental preps and computational models is that they could probe the number of potential variable combinations much deeper than through in vitro experiments alone. The strong similarities between the real-life and computational findings suggest that these oscillatory excitations are consistent, reproducible, and controllable.Triggered activity, which can lead to ventricular arrhythmias and cardiac sudden death, has been largely attributed to sub-cellular phenomena, such as early or delayed afterdepolarizations, and thus to date has largely been studied in isolated single cardiomyocytes. However, these findings have been difficult to translate to tissue and organ-scale experiments, as well-coupled cardiac tissue has notably different electrical properties. This underscores the significance of the study's methodological advances: the use of a constant depolarizing current in a subset of (illuminated) cells to reliably result in triggered activity could facilitate the more consistent evaluation of triggered activity at various scales. An experimental prep that is both repeatable and controllable (i.e., both initiated and terminated through the same means).The authors also substantially explored phase space and single-cell analyses to document how this "hidden" bi-stable phenomenon can be uncovered during emergent collective tissue behavior. Calibration and testing of different aspects (e.g., light intensity, illuminated surface area, electrical pulse frequency, electrical pulse count) and other deeper analyses, as illustrated in Appendix 2, Figures 3-8, are significant and commendable.Given that the study is computational, it is surprising that the authors did not replicate their findings using well-validated adult ventricular cardiomyocyte action potential models, such as ten Tusscher 2006 or O'Hara 2011. This may have felt out of scope, given the nice alignment of rat cardiomyocyte data between in vitro and in silico experiments. However, it would have been helpful peace-of-mind validation, given the significant ionic current differences between neonatal rat and adult ventricular tissue. It is not fully clear whether the pulse trains could have resulted in the same bi-stable oscillatory behavior, given the longer APD of humans relative to rats. The observed phenomenon certainly would be frequency-dependent and would have required tedious calibration for a new cell type, albeit partially mitigated by the relative ease of in silico experiments.For all its strengths, there are likely significant mechanistic differences between this optogenetically tied oscillatory behavior and triggered activity observed in other studies. This is because the constant light-elicited depolarizing current is disrupting the typical resting cardiomyocyte state, thereby altering the balance between depolarizing ionic currents (such as Na+ and Ca2+) and repolarizing ionic currents (such as K+ and Ca2+). The oscillatory excitations appear to later emerge at the border of the illuminated region and non-stimulated surrounding tissue, which is likely an area of high source-sink mismatch. The authors appear to acknowledge differences in this oscillatory behavior and previous sub-cellular triggered activity research in their discussion of ectopic pacemaker activity, which is canonically expected more so from genetic or pathological conditions. Regardless, it is exciting to see new ground being broken in this difficult-to-characterize experimental space, even if the method illustrated here may not necessarily be broadly applicable.

We thank the reviewers for their thoughtful and constructive feedback, as well as for recognizing the conceptual and technical strengths of our work. We are especially pleased that our integrated use of optogenetics, electrical pacing, and computational modelling was seen as a rigorous and innovative approach to investigating spontaneous excitability in cardiac tissue.

At the core of our study was the decision to focus exclusively on neonatal rat ventricular cardiomyocytes. This ensured a tightly controlled and consistent environment across experimental and computational settings, allowing for direct comparison and deeper mechanistic insight. While extending our findings to adult or human cardiomyocytes would enhance translational relevance, such efforts are complicated by the distinct ionic properties and action potential dynamics of these cells, as also noted by Reviewer #2. For this foundational study, we chose to prioritize depth and clarity over breadth.

Our computational domain was designed to faithfully reflect the experimental system. The strong agreement between both domains is encouraging and supports the robustness of our framework. Although some degree of theoretical abstraction was necessary (thereby sometimes making it a bit harder to read), it reflects the intrinsic complexity of the collective behaviours we aimed to capture such as emergent bi-stability. To make these ideas more accessible, we included simplified illustrations, a reduced model, and extensive supplementary material.

A key insight from our work is the emergence of oscillatory behaviour through interaction of illuminated and non-illuminated regions. Rather than replicating classical sub-cellular triggered activity, this behaviour arises from systems-level dynamics shaped by the imposed depolarizing current and surrounding electrotonic environment. By tuning illumination and local pacing parameters, we could reproducibly induce and suppress these oscillations, thereby providing a controllable platform to study ectopy as a manifestation of spatial heterogeneity and collective dynamics.

Altogether, our aim was to build a clear and versatile model system for investigating how spatial structure and pacing influence the conditions under which bistability becomes apparent in cardiac tissue. We believe this platform lays strong groundwork for future extensions into more physiologically and clinically relevant contexts.

In revising the manuscript, we carefully addressed all points raised by the reviewers. We have also responded to each of their specific comments in detail, which are provided below.

**Recommendations for the Authors:**

**Reviewer #1 (Recommendations for the authors):**
Please find my specific comments and suggestions below:(1) Line 64: When first introduced, the concept of 'emergent bi-stability' may not be clear to the reader.

We concur that the full breadth of the concept of emergent bi-stability may not be immediately clear upon first mention. Nonetheless, its components have been introduced separately: “emergent” was linked to multicellular behaviour in line 63, while “bi-stability” was described in detail in lines 39–56. We therefore believe that readers could form an intuitive understanding of the combined term, which will be further clarified as the manuscript develops. To further ease comprehension of the reader, we have added the following clarification to line 64:

“Within this dynamic system of cardiomyocytes, we investigated emergent bi-stability (a concept that will be explained more thoroughly later on) in cell monolayers under the influence of spatial depolarization patterns.”

(2) Lines 67-80: While the introduction until line 66 is extremely well written, the introduction of both cardiac arrhythmia and cardiac optogenetics could be improved. It is especially surprising that miniSOG is first mentioned as a tool for optogenetic depolarisation of cardiomyocytes, as the authors would probably agree that Channelrhodopsins are by far the most commonly applied tools for optogenetic depolarisation (please also refer to the literature by others in this respect). In addition, miniSOG has side effects other than depolarisation, and thus cannot be the tool of choice when not directly studying the effects of oxidative stress or damage.

The reviewer is absolutely correct in noting that channelrhodopsins are the most commonly applied tools for optogenetic depolarisation. We introduced miniSOG primarily for historical context: the effects of specific depolarization patterns on collective pacemaker activity were first observed with this tool (Teplenin et al., 2018). In that paper, we also reported ultralong action potentials, occurring as a side effect of cumulative miniSOG-induced ROS damage. In the following paragraph (starting at line 81), we emphasize that membrane potential can be controlled much better using channelrhodopsins, which is why we employed them in the present study.

(3) Line 78: I appreciate the concept of 'high curvature', but please always state which parameter(s) you are referring to (membrane voltage in space/time, etc?).

We corrected our statement to include the specification of space curvature of the depolarised region:

“In such a system, it was previously observed that spatiotemporal illumination can give rise to collective behaviour and ectopic waves (Teplenin et al. (2018)) originating from illuminated/depolarised regions (with high spatial curvature).”

(4) Line 79: 'bi-stable state' - not yet properly introduced in this context.

The bi-stability mentioned here refers back to single cell bistability introduced in Teplenin et al. (2018), which we cited again for clarity.

“These waves resulted from the interplay between the diffusion current and the single cell bi-stable state (Teplenin et al. (2018)) that was induced in the illuminated region.”

(5) Line 84-85: 'these ion channels allow the cells to respond' - please describe the channel used; and please correct: the channels respond to light, not the cells. Re-ordering this paragraph may help, because first you introduce channels for depolarization, then you go back to both de- and hyperpolarization. On the same note, which channels can be used for hyperpolarization of cardiomyocytes? I am not aware of any, even WiChR shows depolarizing effects in cardiomyocytes during prolonged activation (Vierock et al. 2022). Please delete: 'through a direct pathway' (Channelrhodopsins a directly light-gated channels, there are no pathways involved).

We realised that the confusion arose from our use of incorrect terminology: we mistakenly wrote hyperpolarisation instead of repolarisation. In addition to channelrhodopsins such as WiChR, other tools can also induce a repolarising effect, including light-activatable chloride pumps (e.g., JAWS). However, to improve clarity, we recognize that repolarisation is not relevant to our manuscript and therefore decided to remove its mention (see below). Regarding the reported depolarising effects of WiChR in Vierock et al. (2022), we speculate that these may arise either from the specific phenotype of the cardiomyocytes used in the study, i.e. human induced pluripotent stem cell-derived atrial myocytes (aCMs), or from the particular ionic conditions applied during patch-clamp recordings (e.g., a bath solution containing 1 mM KCl). Notably, even after prolonged WiChR activation, the aCMs maintained a strongly negative maximum diastolic potential of approximately –55 mV.

“Although effects of illuminating miniSOG with light might lead to formation of depolarised areas, it is difficult to control the process precisely since it depolarises cardiomyocytes indirectly. Therefore, in this manuscript, we used light-sensitive ion channels to obtain more refined control over cardiomyocyte depolarisation. These ion channels allow the cells to respond to specific wavelengths of light, facilitating direct depolarisation (Ördög et al. (2021, 2023)). By inducing cardiomyocyte depolarisation only in the illuminated areas, optogenetics enables precise spatiotemporal control of cardiac excitability, an attribute we exploit in this manuscript (Appendix 2 Figure 1).”

(6) Figure 1: What would be the y-axis of the 'energy-like curves' in B? What exactly did you plot here?

The graphs in Figure 1B are schematic representations intended to clarify the phenomenon for the reader. They do not depict actual data from any simulation or experiment. We clarified this misunderstanding by specifying that Figure 1B is a schematic representation of the effects at play in this paper.

“(B) Schematic representation showing how light intensity influences collective behaviour of excitable systems, transitioning between a stationary state (STA) at low illumination intensities and an oscillatory state (OSC) at high illumination intensities. Bi-stability occurs at intermediate light intensities, where transitions between states are dependent on periodic wave train properties. TR. OSC, transient oscillations.”

To expand slightly beyond the paper: our schematic representation was inspired by a common visualization in dynamical systems used to illustrate bi-stability (for an example, see Fig. 3 in Schleimer, J. H., Hesse, J., Contreras, S. A., & Schreiber, S. (2021). Firing statistics in the bistable regime of neurons with homoclinic spike generation. Physical Review E, 103(1), 012407.). In this framework, the y-axis can indeed be interpreted as an energy landscape, which is related to a probability measure through the Boltzmann distribution: \begin{document}$E=\beta \ln \left(\frac{1}{p}\right)$\end{document}. Here, p denotes the probability of occupying a particular state (STA or OSC). This probability can be estimated from the area (BCL × number of pulses) falling within each state, as shown in Fig. 4C. Since an attractor corresponds to a high-probability state, it naturally appears as a potential well in the landscape.

(7) Lines 92-93: 'this transition resulted for the interaction of an illuminated region with depolarized CM and an external wave train' - please consider rephrasing (it is not the region interacting with depolarized CM; and the external wave train could be explained more clearly).

We rephrased our unclear sentence as follows:

“This transition resulted from the interaction of depolarized cardiomyocytes in an illuminated region with an external wave train not originating from within the illuminated region.”

(8) Figure 2 and elsewhere: When mentioning 'frequency', please state frequency values and not cycle lengths. Please also reconsider your distinction between high and low frequencies; 200 ms (5 Hz) is actually the normal heart rate for neonatal rats (300 bpm).

In the revised version, we have clarified frequency values explicitly and included them alongside period values wherever frequency is mentioned, to avoid any ambiguity. We also emphasize that our use of "high" and "low" frequency is strictly a relative distinction within the context of our data, and not meant to imply a biological interpretation.

(9) Lines 129-131: Why not record optical maps? Voltage dynamics in the transition zone between depolarised and non-depolarised regions might be especially interesting to look at?

We would like to clarify that optical maps were recorded for every experiment, and all experimental traces of cardiac monolayer activity were derived from these maps. We agree with the reviewer that the voltage dynamics in the transition zone are particularly interesting. However, we selected the data representations that, in our view, best highlight the main mechanisms. When we analysed full voltage profiles, they didn’t add extra insights to this main mechanism. As the other reviewer noted, the manuscript already presents a wide range of regimes, so we decided not to introduce further complexity.

(10) Lines 156-157: Why was the model not adapted to match the biophysical properties (e.g., kinetics, ion selectivity, light sensitivity) of Cheriff?

The model was not adapted to the biophysical properties of Cheriff, because this would entail a whole new study involving extensive patch-clamping experiments, fitting, and calibration to model the correct properties of the ion channel. Beyond considerations of time efficiency, incorporating more specific modelling parameters would not change the essence of our findings. While numeric parameter ranges might shift, the core results would remain unchanged. This is a result of our experimental design where we applied constant illumination of long duration (6s or longer), thus making a difference in kinetical properties of an optogenetic tool irrelevant. In addition, we were able to observe qualitatively similar phenomena using many other depolarising optogenetic tools (e.g. ChR2, ReaChR, CatCh and more) in our *in-vitro experiments.* We ended up with Cheriff as our optotool-of-choice for the practical reasons of good light-sensitivity and a non-overlapping spectrum with our fluorescent dyes.

Therefore, computationally using a more general depolarising ion channel hints at the more general applicability of the observed phenomena, supporting our claim of a universal mechanism (demonstrated experimentally with CheRiff and computationally with ChR2).

(11) Line 158: 1.7124 mW/mm^2 - While I understand that this is the specific intensity used as input in the model, I am convinced that the model is not as accurate to predict behaviour at this specific intensity (4 digits after the comma), especially given that the model has not been adapted to Cheriff (probably more light sensitive than ChR2). Can this be rephrased?

We did not aim for quantitative correspondence between the computational model and the biological experiments, but rather for qualitative agreement and mechanistic insight (see line 157). Qualitative comparisons are computationally obtained in a whole range of different intensities, as demonstrated in the 3D diagram of Fig. 4C. We wanted to demonstrate that at one fixed light intensity (chosen to be 1.7124 mW/mm^2 for the most clear effect), it was possible for all three states (STA, OSC. TR. OSC.) to coexist depending on the number of pulses and their period. Therefore the specific intensity used in the computational model is correct, and for reproducibility, we have left it unchanged while clarifying that it refers specifically to the *in silico* model:

“Simulating at a fixed constant illumination of 1.7124 𝑚𝑊∕𝑚𝑚^2^ and a fixed number of 4 pulses, frequency dependency of collective bi-stability was reproduced in Figure 4A.”

(12) Lines 160, 165, and elsewhere: 'Once again, Once more' - please delete or rephrase.

We agree that we could have written these binding words better and reformulated them to:

“Similar to the experimental observations, only intermediate electrical pacing frequencies (500-𝑚𝑠 period) caused transitions from collective stationary behaviour to collective oscillatory behaviour and ectopic pacemaker activity had periods (710 𝑚𝑠) that were different from the stimulation train period (500 𝑚𝑠). Figure 4B shows the accumulation of pulses necessary to invoke a transition from the collective stationary state to the collective oscillatory state at a fixed stimulation period (600 𝑚𝑠). Also in the in silico simulations, ectopic pacemaker activity had periods (750 𝑚𝑠) that were different from the stimulation train period (600 𝑚𝑠). Also for the transient oscillatory state, the simulations show frequency selectivity (Appendix 2 Figure 4B).”

(13) Line 171: 'illumination strength': please refer to 'light intensity'.

We have revised our formulation to now refer specifically to “light intensity”:

“We previously identified three important parameters influencing such transitions: light intensity, number of pulses, and frequency of pulses.”

(14) Lines 187-188: 'the illuminated region settles into this period of sending out pulses' - please rephrase, the meaning is not clear.

We reformulated our sentence to make its content more clear to the reader:

“For the conditions that resulted in stable oscillations, the green vertical lines in the middle and right slices represent the natural pacemaker frequency in the oscillatory state. After the transition from the stationary towards the oscillatory state, oscillatory pulses emerging from the illuminated region gradually dampen and stabilize at this period, corresponding to the natural pacemaker frequency.”

(15) Figure 7: (A)- please state in the legend which parameter is plotted on the y-axis (it is included in the main text, but should be provided here as well); (C) The numbers provided in brackets are confusing. Why is (4) a high pulse number and (3) a low pulse number? Why not just state the number of pulses and add alpha, beta, gamma, and delta for the panels in brackets? I suggest providing the parameters (e.g., 800 ms cycle length, 2 pulses, etc) for all combinations, but not rate them with low, high, etc. (see also comment above).

We appreciate the reviewer’s comments and have revised the caption for figure 7, which now reads as follows:

“Figure 7. Phase plane projections of pulse-dependent collective state transitions. (A) Phase space trajectories (displayed in the Voltage – x_r_ plane) of the NRVM computational model show a limit cycle (OSC) that is not lying around a stable fixed point (STA). (B) Parameter space slice showing the relationship between stimulation period and number of pulses for a fixed illumination intensity (1.72 𝑚𝑊 ∕𝑚𝑚2) and size of the illuminated area (67 pixels edge length). Letters correspond to the graphs shown in C. (C) Phase space trajectories for different combinations of stimulus train period and number of pulses (α: 800 ms cycle length + 2 pulses, β: 800 ms cycle length + 4 pulses, γ: 250 ms cycle length + 3 pulses, δ: 250 ms cycle length + 8 pulses). α and δ do not result in a transition from the resting state to ectopic pacemaker activity, as under these circumstances the system moves towards the stationary stable fixed point from outside and inside the stable limit cycle, respectively. However, for β and γ, the stable limit cycle is approached from outside and inside, respectively, and ectopic pacemaker activity is induced.”

(16) Line 258: 'other dimensions by the electrotonic current' - not clear, please rephrase and explain.

We realized that our explanation was somewhat convoluted and have therefore changed the text as follows:

“Rather than producing oscillations, the system returns to the stationary state along dimensions other than those shown in Figure 7C (Voltage and x_r_), as evidenced by the phase space trajectory crossing itself. This return is mediated by the electrotonic current.”

(17) Line 263: ‘increased too much’ – please rephrase using scientific terminology.

We rephrased our sentence to:

“However, this is not a Hopf bifurcation, because in that case the system would not return to the stationary state when the number of pulses exceeds a critical threshold.”

(18) Line 275: 'stronger diffusion/electrotonic influence from the non-illuminated region' - not sure diffusion is the correct term here. Please explain by taking into account the membrane potential. Please make sure to use proper terminology. The same applies to lines 281-282.

We appreciate this comment, which prompted us to revisit on our text. We realised that some sections could be worded more clearly, and we also identified an error in the legend of Supplementary Figure 7. The corresponding corrections are provided below:

“However, repolarisation reserve does have an influence, prolonging the transition when it is reduced (Appendix 2 Figure 7). This effect can be observed either by moving further from the boundary of the illuminated region, where the electrotonic influence from the non-illuminated region is weaker, or by introducing ionic changes, such as a reduction in I_Ks_ and/or I_to_. For example, because the electrotonic influence is weaker in the center of the illuminated region, the voltage there is not pulled down toward the resting membrane potential as quickly as in cells at the border of the illuminated zone.”

“To add a multicellular component to our single cell model we introduced a current that replicates the effect of cell coupling and its associated electrotonic influence.”

“Figure 7. The effect of ionic changes on the termination of pacemaker activity. The mechanism that moves the oscillating illuminated tissue back to the stationary state after high frequency pacing is dependent on the ionic properties of the tissue, i.e. lower repolarisation reserves (20% 𝐼_𝐾𝑠_ + 50% 𝐼_𝑡𝑜_) are associated with longer transition times.”

(19) Line 289: -58 mV (to be corrected), -20 mV, and +50 mV - please justify the selection of parameters chosen. This also applies elsewhere- the selection of parameters seems quite arbitrary, please make sure the selection process is more transparent to the reader.

Our choice of parameters was guided by the dynamical properties of the illuminated cells as well as by illustrative purposes. The value of –58 mV corresponds to the stimulation threshold of the model. The values of 50 mV and –20 mV match those used for single-cell stimulation (Figure 8C2, right panel), producing excitable and bistable dynamics, respectively. We refer to this point in line 288 with the phrase “building on this result.” To maintain conciseness, we did not elaborate on the underlying reasoning within the manuscript and instead reported only the results.

We also corrected the previously missed minus sign: -58 mV.

(20) Figure 8 and corresponding text: I don't understand what stimulation with a voltage means. Is this an externally applied electric field? Or did you inject a current necessary to change the membrane voltage by this value? Please explain.

Stimulation with a specific voltage is a standard computational technique and can be likened to performing a voltage-clamp experiment on each individual cell. In this approach, the voltage of every cell in the tissue is briefly forced to a defined value.

(21) Figure 8C- panel 2: Traces at -20 mV and + 50 mV are identical. Is this correct? Please explain.

Yes, that is correct. The cell responds similarly to a voltage stimulus of -20 mV or one of 50 mV, because both values are well above the excitation threshold of a cardiomyocyte.

(22) Line 344 and elsewhere: 'diffusion current' - This is probably not the correct terminology for gap-junction mediated currents. Please rephrase.

A diffusion current is a mathematical formulation for a gap junction mediated current here, so , depending on the background of the reader, one of the terms might be used focusing on different aspects of the results. In a mathematical modelling context one often refers to a diffusion current because cardiomyocytes monolayers and tissues can be modelled using a reaction-diffusion equation. From the context of fine-grain biological and biophysical details, one uses the term gap-junction mediated current. Our choice is motivated by the main target audience we have in mind, namely interdisciplinary researchers with a core background in the mathematics/physics/computer science fields.

However, to not exclude our secondary target audience of biological and medical readers we now clarified the terminology, drawing the parallel between the different fields of study at line 79:

“These waves resulted from the interplay between the diffusion current (also known in biology/biophysics as the gap junction mediated current) and the bi-stable state that was induced in the illuminated region.”

(23) Lines 357-58: 'Such ectopic sources are typically initiated by high frequency pacing' - While this might be true during clinical testing, how would you explain this when not externally imposed? What could be biological high-frequency triggers?

Biological high-frequency triggers could include sudden increases in heart rates, such as those induced by physical activity or emotional stress. Another possibility is the occurrence of paroxysmal atrial or ventricular fibrillation, which could then give rise to an ectopic source.

(24) Lines 419-420: 'large ionic cell currents and small repolarising coupling currents'. Are coupling currents actually small in comparison to cellular currents? Can you provide relative numbers (~ratio)?

Coupling currents are indeed small compared to cellular currents. This can be inferred from the I-V curve shown in Figure 8C1, which dips below 0 and creates bi-stability only because of the small coupling current. If the coupling current were larger, the system would revert to a monostable regime. To make this more concrete, we have now provided the exact value of the coupling current used in Figure 8C1.

“Otherwise, if the hills and dips of the N-shaped steady-state IV curve were large (Figure 8C-1), they would have similar magnitudes as the large currents of fast ion channels, preventing the subtle interaction between these strong ionic cell currents and the small repolarising coupling currents (-0.103649 ≈ 0.1 pA).”

(25) Line 426: Please explain how ‘voltage shocks’ were modelled.

We would like to refer the reviewer to our response to comment (20) regarding how we model voltage shocks. In the context of line 426, a typical voltage shock corresponds to a tissue-wide stimulus of 50 mV. Independent of our computational model, line 426 also cites other publications showing that, in clinical settings, high-voltage shocks are unable to terminate ectopic sustained activity, consistent with our findings.

(26) Lines 429 ff: 0.2pA/pF would correspond to 20 pA for a small cardiomyocyte of 100 pF, this current should be measurable using patch-clamp recordings.

In trying to be succinct, we may have caused some confusion. The difference between the dips (*≈*-0.07 pA/pF) and hills (_≈_0.11 pA/pF) is approximately 0.18 pA/pF. For a small cardiomyocyte, this corresponds to deviations from zero of roughly ±10 pA. Considering that typical RMS noise levels in whole-cell patch-clamp recordings range from 2-10 pA , it is understandable that detecting these peaks and dips in an I-V curve (average current after holding a voltage for an extended period) is difficult. Achieving statistical significance would therefore require patching a large number of cells.

Given the already extensive scope of our manuscript in terms of techniques and concepts, we decided not to pursue these additional patch-clamp experiments.

**Reviewer #2 (Recommendations for the authors):**
Given the deluge of conditions to consider, there are several areas of improvement possible in communicating the authors' findings. I have the following suggestions to improve the manuscript.(1) Please change "pulse train" straight pink bar OR add stimulation marks (such as "*", or individual pulse icons) to provide better visual clarity that the applied stimuli are "short ON, long OFF" electrical pulses. I had significant initial difficulty understanding what the pulse bars represented in Figures 2, 3, 4A-B, etc. This may be partially because stimuli here could be either light (either continuous or pulsed) or electrical (likely pulsed only). To me, a solid & unbroken line intuitively denotes a continuous stimulation. I understand now that the pink bar represents the entire pulse-train duration, but I think readers would be better served with an improvement to this indicator in some fashion. For instance, the "phases" were much clearer in Figures 7C and 8D because of how colour was used on the Vm(t) traces. (How you implement this is up to you, though!)

We have addressed the reviewer’s concern and updated the figures by marking each external pulse with a small vertical line (see below).

(2) Please label the electrical stimulation location (akin to the labelled stimulation marker in circle 2 state in Figure 1A) in at least Figures 2 and 4A, and at most throughout the manuscript. It is unclear which "edge" or "pixel" the pulse-train is originating from, although I've assumed it's the left edge of the 2D tissue (both in vitro and silico). This would help readers compare the relative timing of dark blue vs. orange optical signal tracings and to understand how the activation wavefront transverses the tissue.

We indicated the pacing electrode in the optical voltage recordings with a grey asterisk. For the *in silico* simulations, the electrode was assumed to be far away, and the excitation was modelled as a parallel wave originating from the top boundary, indicated with a grey zone.

(3) Given the prevalence of computational experiments in this study, I suggest considering making a straightforward video demonstrating basic examples of STA, OSC, and TR.OSC states. I believe that a video visualizing these states would be visually clarifying to and greatly appreciated by readers. Appendix 2 Figure 3 would be the no-motion visualization of the examples I'm thinking of (i.e., a corresponding stitched video could be generated for this). However, this video-generation comment is a suggestion and not a request.

We have included a video showing all relevant states, which is now part of the Supplementary Material.

(4) Please fix several typos that I found in the manuscript:

(4A) Line 279: a comma is needed after i.e. when used in: "peculiar, i.e. a standard". However, this is possibly stylistic (discard suggestion if you are consistent in the manuscript).

(4B) Line 382: extra period before "(Figure 3C)".

(4C) Line 501: two periods at end of sentence "scientific purposes.." .

We would like to thank the reviewer for pointing out these typos. We have corrected them and conducted an additional check throughout the manuscript for minor errors.